# Missing Data Imputation and Acquisition with Deep Hierarchical Models and Hamiltonian Monte Carlo

**Ignacio Peis**
Universidad Carlos III de Madrid
Madrid, Spain
ipeis@tsc.uc3m.es

**Chao Ma**
Microsoft Research
University of Cambridge
Cambridge, UK
cm905@cam.ac.uk

**José Miguel Hernández-Lobato**
University of Cambridge
Cambridge, UK
jmh233@cam.ac.uk

## Abstract

Variational Autoencoders (VAEs) have recently been highly successful at imputing and acquiring heterogeneous missing data. However, within this specific application domain, existing VAE methods are restricted by using only one layer of latent variables and strictly Gaussian posterior approximations. To address these limitations, we present HH-VAEM, a Hierarchical VAE model for mixed-type incomplete data that uses Hamiltonian Monte Carlo with automatic hyper-parameter tuning for improved approximate inference. Our experiments show that HH-VAEM outperforms existing baselines in the tasks of missing data imputation and supervised learning with missing features. Finally, we also present a sampling-based approach for efficiently computing the information gain when missing features are to be acquired with HH-VAEM. Our experiments show that this sampling-based approach is superior to alternatives based on Gaussian approximations.

## 1 Introduction

Many real-world unsupervised learning tasks require dealing with complicated datasets with mixed types (real, positive-valued, continuous, or discrete) and missing values. For this purpose, variational autoencoders [24, 25, 41] stand out in the recent literature as robust generative models that efficiently handle high-dimensional data. However, in their naive configuration, every data dimension is assumed to have similar statistical properties (i.e., homogeneity), and all dimensions are considered to be completely observed. Both assumptions won't hold in many real-world scenarios. Recent works have adapted VAEs to handle incomplete [9, 14, 34, 36] and mixed-type data [18, 33, 38], and demonstrated improved performance in downstream tasks such as missing data imputation and active information acquisition. Despite these advances, existing approaches are far from optimal as they are based on restrictive design choices: 1), only one layer of latent variables are considered; 2), Gaussian posterior approximations are usually adopted. These will lead to limited flexibility and additional bias, especially under real-world settings with complex mixed-type incomplete data.

In the literature, the issue of model flexibility and inference bias are often addressed separately. For example, approximate inference bias can be reduced by using Monte Carlo sampling [46, 50]. More specifically, Hamiltonian Monte Carlo (HMC) [5, 12] stands out among MCMC methods in machine learning due to its superior efficiency for exploring the target density. In the context of VAE, HMC has also been combined with stochastic variational inference [7] for improving the training of VAEs.

36th Conference on Neural Information Processing Systems (NeurIPS 2022).

On the other hand, the flexibility of VAEs can be improved by considering hierarchical VAEs with multiple layers of hidden variables [8, 35, 48, 52]. By using a hierarchical structure in the latent space, they enforce the information to flow from high-level representations to more specific observable factors, imitating the way information is often organized in the real world.

However, the issue of modeling flexibility and approximate inference bias are often heavily intertwined, and addressing them simultaneously in a *joint* manner is highly non-trivial. The hierarchical organization of the latent variables creates complicated posterior dependencies that are not straightforward to deal with and require special consideration. To improve Gaussian approximate inference, most works opt by defining shared paths between the recognition and generative networks. While this makes hierarchical VAEs practical, the bias introduced by the Gaussian approximations is still present. To the best of our knowledge, none of the aforementioned hierarchical VAEs has been previously combined with Monte Carlo algorithms for improving over standard Gaussian approximate inference.

To overcome such limitations, we focus on training new hierarchical VAE models for heterogeneous mixed-type data with HMC. Our models can be used for missing data imputation and for supervised learning with missing data. We also present a sampling-based framework that allows our models to perform accurate sequential active information acquisition.

Our main contributions are as follows:

- We present HH-VAEM, a deep hierarchical model for handling mixed-type incomplete data that uses HMC with automatic hyper-parameter tuning for outperforming amortized variational inference by generating low bias samples from the true posterior.

- We propose a sampling-based strategy for missing feature acquisition that benefits from the improved inference of HH-VAEM. By using histograms to estimate the mutual information, this strategy achieves lower bias than other Gaussian-based alternatives.

- We exhaustively evaluate HH-VAEM in the tasks of 1) missing data imputation, 2) supervised learning with missing data and 3) information acquisition with our sampling-based strategy. In all cases we report significant gains with respect to baselines.

## 2 Related work

### 2.1 VAEs for mixed-type incomplete data

Variational Autoencoders [24, 41] are deep generative models that make use of encoder and decoder networks for mapping data into a latent Gaussian distribution, and reconstructing the latent codes into the original observational space, respectively. The parameters of these networks are trained using amortized Variational Inference [10, 55] optimizing a lower bound (ELBO) on the log evidence: $\mathbb{E}_{q_{\boldsymbol{z}}(\boldsymbol{z}|\boldsymbol{x})} \log \frac{p_\theta(\boldsymbol{x},\boldsymbol{z})}{q_\psi(\boldsymbol{z}|\boldsymbol{x})}$, where the generative model $p_\theta(\boldsymbol{x},\boldsymbol{z})$ can be expressed in terms of the likelihood $p_\theta(\boldsymbol{x}|\boldsymbol{z})$ and the prior $p(\boldsymbol{z})$. In mixed-type data, the vector $\boldsymbol{x}$ is composed by data from different types: real, positive real, categorical, binary, etc. A naive approach is to consider a factorized decoder using different likelihood contributions $p_\theta(\boldsymbol{x}|\boldsymbol{z}) = \prod_d p_\theta(x_d|\boldsymbol{z})$ [1, 38]. Nonetheless, the problem of handling unbalanced likelihoods leads to the domination of some dimensions during the optimization process. In [33], authors propose a solution using a set of *marginal* VAEs that encode each feature into a Gaussian uni-dimensional space, and a *dependency* VAE that captures the inter-dimensional dependencies more effectively using balanced Gaussian likelihoods.

By marginalizing each dimension of the decoder, incomplete data can be easily handled by dividing the vector $\boldsymbol{x}$ into the observed $\boldsymbol{x}_O$ and unobserved $\boldsymbol{x}_U$ parts. This methodology is completely valid when using the missing-at-random (MAR) assumption [30], i.e. assuming the missing mechanism is independent of the missing values. In this our work, we use the same assumption. As proposed in [38] and [36], the ELBO objective is transformed into a lower bound on the observed data, and the unobserved data is replaced with zeros.

### 2.2 Hierarchical VAEs

Hierarchical models have been successfully employed in deep generative modeling, [2, 44, 45]. In VAEs, defining a hierarchical latent space for VAEs can be straightforward. Nevertheless, potential pitfalls require special attention. Concretely, if the decoder is powerful enough, the model tends

to uniquely use the shallowest layers, ignoring the deepest ones and falling into the well-known problem of *posterior collapse* [35, 40, 53]. In the last few years, several works have investigated possible hierarchical structures for VAEs. In [48], a bottom-up deterministic path is used along with a top-down inference path that shares parameters with the top-down structure of the generative model. In [35], the authors use a bidirectional stochastic inference path. More recently, [52] or [8] have adapted these architectures to complex datasets and high quality images. Possibly motivated by the residual connections in [23], all these works use a shared path between recognition and generative models that helps in tying the divergences between approximations and priors in the ELBO.

## 2.3 Hamiltonian Monte Carlo

HMC [4, 12, 39] is a particularly effective MCMC algorithm for sampling from a target distribution $p(z) = \frac{1}{\mathcal{Z}} p^*(z)$ where $\mathcal{Z}$ is the normalization constant and $z$ is a $d$-dimensional vector. By augmenting this model to $p(r, z) = \mathcal{N}(r; 0, M) p(z)$, and denoting $r$ as the *momentum* variable with diagonal covariance matrix $M$, with the same dimensionality as $z$, HMC samples are obtained from the distribution by simulating the time-evolution of a fictitious physical system.

The algorithm starts by firstly sampling $z$ and $r$ from an initial proposal and the momentum distribution, respectively. Chains with length $T$ are built by recurrently proposing and accepting new states. To propose a new state, the Hamiltonian dynamics are simulated using a symplectic integrator, Leapfrog being the most common choice. The following updates are repeated for $l = 1 : LF$ steps:

$$
\begin{aligned}
r_{l+\frac{1}{2}} &= r_l + \frac{1}{2} \phi \odot \nabla_{z_l} \log p^*(z_l), \\
z_{l+1} &= z_l + r_{l+\frac{1}{2}} \odot \phi \odot \frac{1}{M}, \\
r_{l+1} &= r_{l+\frac{1}{2}} + \frac{1}{2} \phi \odot \nabla_{z_{l+1}} \log p^*(z_{l+1}),
\end{aligned}
\tag{1}
$$

where $\odot$ refers to the Hadamard product, and $\phi$ is the *step size* hyperparameter. Although it is typically defined as a scalar for simplicity, a $d$-dimensional vector can be considered to apply different step sizes per dimension, or further, as considered in this work, a $T \times d$ matrix to apply different steps per each proposal of the chain. The new proposal $(z', r')$ is accepted with probability $min[1, \exp(-H(z', r') + H(z, r))]$, where

$$
H(z, r) = -\log p^*(z) + \frac{1}{2} r^T M^{-1} r.
\tag{2}
$$

For the consecutive $T$ proposals, a new momentum $r$ is resampled and the updates of eq. (1) are repeated for $LF$ steps to update the state if $(z', r')$ is accepted.

## 2.4 Active Feature Acquisition

Among all the Active Learning techniques, Active Feature Acquisition [20, 37, 43, 49] is of special interest in cost-sensitive applications for modeling a trade-off between the improvement of predictions and the cost of acquiring new data at the feature level. Several works in the recent literature have studied methods for performing the task of sequentially acquiring high-value information by selecting features that maximize an information theoretical reward function and enhance the accuracy of the predictions. This task is denoted by SAIA (Sequential Active Information Acquisition). In [34], an efficient method is proposed for approximating a non-tractable reward by using the encoder of a VAE that handles missing data. In [33] they extend this method for handling mixed-type data. Both works estimate the reward by relying on Gaussian approximations given by the encoder networks.

## 3 Hamiltonian hierarchical VAE for mixed-type incomplete data (HH-VAEM)

The HH-VAEM model (Figure 1 (a)) is a Hierarchical VAE for mixed-type, incomplete data that incorporates HMC with automatic hyper-parameter optimization for sampling from the posterior of the latent variables. In a first stage, the mixed-type data is encoded into marginal Gaussian posterior distributions as given by univariate VAEs fitted to each data dimension. In a second stage, a hierarchical structure captures the dependencies among the standardized, homogeneous dimensions with well-balanced Gaussian likelihoods. The model is trained using samples from the posterior of the

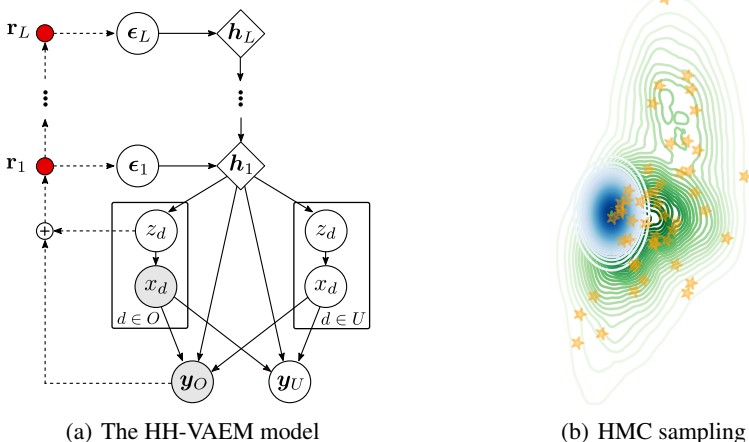

(a) The HH-VAEM model       (b) HMC sampling

Figure 1: The HH-VAEM model (a). Illustrative example (b): samples $\boldsymbol{\epsilon}^{(T)}$ obtained with HMC (orange) following the true posterior $p(\boldsymbol{\epsilon}|\boldsymbol{x}_O, \boldsymbol{y}_O)$ (green) using the Gaussian distribution given by the encoder $q^{(0)}(\boldsymbol{\epsilon}|\boldsymbol{x}_O, \boldsymbol{y}_O)$ (blue) as the initial proposal, with latent dimension $M = 2$.

hierarchical latent variables by means of HMC, whilst the HMC hyper-parameters are automatically tuned. A more detailed description is provided in the following subsections.

## 3.1 Notation

The model generates both data $\boldsymbol{x} \in \mathbb{R}^D$ and output $\boldsymbol{y} \in \mathbb{R}^P$, where each of these variables is divided into observed parts $\boldsymbol{x}_O$, $\boldsymbol{y}_O$ and unobserved parts $\boldsymbol{x}_U$, $\boldsymbol{y}_U$. Each dimension of $\boldsymbol{x}$ is denoted by $x_d$. A training set is composed of $N$ observations as tuples $(\boldsymbol{x}_O^{(n)}, \boldsymbol{y}_O^{(n)})$. For ease of notation, we omit the observation index $n$, and the objectives are presented for a single observation. The *dependency* latent space is composed of $L$ latent variables $[\boldsymbol{\epsilon}_1, ..., \boldsymbol{\epsilon}_L]$, with $\boldsymbol{\epsilon}_l \in \mathbb{R}^{m_l}$. The dimension of the joint latent distribution is $\sum_l m_l = M$. In the marginal VAEs, the latent variables $z_d$ are unidimensional.

## 3.2 Handling heterogeneous incomplete data

Following the strategy proposed by [33], we perform a two-staged approach for handling heterogeneous data. The marginal distribution of each feature $p_{\theta_d}(x_d)$ is modeled by a one-dimensional VAE. First, the $D$ marginal VAEs are trained independently by maximizing the marginal ELBO over observed points

$$\mathcal{L}_d(x_d; \{\theta_d, \gamma_d\}) = \mathbb{1}(x_d \in \boldsymbol{x}_O)\mathbb{E}_{q_{\gamma_d}(z_d|x_d)} \log \frac{p_{\theta_d}(x_d, z_d)}{q_{\gamma_d}(z_d|x_d)}, \tag{3}$$

where $\mathbb{1}(x_d \in \boldsymbol{x}_O)$ is an indicator function that activates the ELBO when the feature is observed. Under the missing-at-random assumption, which is the one considered in this work, Equation (3) leads to a lower bound on observed data likelihoods. Second, a *dependency* VAE encodes a vector $\boldsymbol{z}$ with the concatenated samples from the marginal posteriors $q_{\gamma_d}(z_d|x_d)$ into the global latent variable $\boldsymbol{h}$, using zero-filling for the unobserved variables. By using this approach, $\boldsymbol{z}$ is now homogeneous and can then be easily modeled using a standard Gaussian decoder. By contrast, other works [13, 38] directly operate with different decoding likelihoods per data type. This approach often leads to having very different magnitudes in the ELBO and may reduce learning efficiency. The ELBO for the second stage dependency VAE is

$$\mathcal{L}(\boldsymbol{x}_O, \boldsymbol{y}_O; \{\theta, \psi\}) = \mathbb{E}_{q_\psi} \left[ \log \frac{p_\theta(\boldsymbol{z}_O, \boldsymbol{y}_O, \boldsymbol{\epsilon})}{q_\psi(\boldsymbol{\epsilon}|\boldsymbol{z}_O, \boldsymbol{x}_O, \boldsymbol{y}_O)} \right] \tag{4}$$

where $\boldsymbol{\epsilon} = \{\boldsymbol{\epsilon}_1, ..., \boldsymbol{\epsilon}_L\}$ is a set of reparameterized hierarchical latent variables. Further details on the design of the hierarchical dependency VAE are provided below.

### 3.3 Predictive enhancement

The combination of generative and discriminative models is an effective well-studied strategy for dealing with predictive models under missing data [15, 51]. In [15], they model $p(\boldsymbol{x})$ for imputing missing data using a Gaussian Mixture Model. In a deep learning context, recent supervised VAE models have revisited this combination [21, 47] or used factorisations of type $p(\boldsymbol{z})p(\boldsymbol{x}|\boldsymbol{z})p(\boldsymbol{y}|\boldsymbol{z})$ [22] to learn meaningful representations. In [28] the authors propose a deep generative model with factorisation $p(\boldsymbol{z})p(\boldsymbol{x}|\boldsymbol{z})p(\boldsymbol{y}|\boldsymbol{x}, \boldsymbol{z})$ for detecting adversarial attacks.

With the aim at reinforcing the prediction of the variable of interest, we turn into a supervised model by including a separate predictor for $p_{\theta_y}(\boldsymbol{y}|\hat{\boldsymbol{x}}, \boldsymbol{h})$, apart from the decoder $p_{\theta_z}(\boldsymbol{z}|\boldsymbol{h})$. The vector $\hat{\boldsymbol{x}} = (x_i \in \boldsymbol{x}_O, \ \hat{x}_j \in \boldsymbol{x}_U)$ includes the observed part and imputation of the missing variables $\hat{x}_j$ by decoding the latent $\boldsymbol{h}$ into $\boldsymbol{z}$ using $p(\boldsymbol{z}|\boldsymbol{h_1})$, and each dimension $z_j \in \boldsymbol{z}$ into $\hat{x}_j$ using $p(x_j|z_j)$. The predictor parameters $\theta_y$ are optimized along with the decoder parameters $\theta_z$.

### 3.4 Hierarchical reparameterized latent space

A hierarchical structure over the latent space $\boldsymbol{h} = \{\boldsymbol{h}_1, ..., \boldsymbol{h}_L\}$ enriches the prior assumptions and permits a flexible generation of data in a more natural fashion. Nevertheless, as stated in [4, 5], HMC can be pathological when used for sampling from hierarchical densities, where the magnitude of autoregressive variations increase with the depth. For approximating the Hamiltonian dynamics, inside each Leapfrog integrator step, gradients $\nabla_{\boldsymbol{h}_{1:L}} \log p^*(\boldsymbol{h}_{1:L})$ are required. Due to the strong curvature regions, huge norm of high-order derivatives are backpropagated and might eventually explode, ending in overflow issues (Figure 35 in [5]).

If we were to run our HMC method over the hierarchical variables without any reparameterization (Figure 4a), by the time the states reached the aforementioned problematic regions, the integrator would diverge and we would experience the aforementioned overflow problems. By rejecting these problematic states, chains would get stuck close to the proposal and the hierarchical density would not be properly explored (concluding that HMC would not improve the Gaussian proposal). To give an example, in [52], the AR path $p(\boldsymbol{h}_l|\boldsymbol{h}_{<l})$ would provoke instabilities inside the HMC integrator due to huge gradients in $\nabla_{\boldsymbol{h}_{1:L}} \log p^*(\boldsymbol{h}_{1:L})$.

We successfully solved this issue by introducing a hierarchical reparameterization technique. The representation at each layer is reparameterized from variable $\boldsymbol{\epsilon}_l$ with standard Gaussian prior $p(\boldsymbol{\epsilon}_l)$

$$\boldsymbol{h}_l = f_{\mu_l}(\boldsymbol{h}_{l+1}) + f_{\sigma_l}(\boldsymbol{h}_{l+1}) \cdot \boldsymbol{\epsilon}_l, \tag{5}$$

where the functions $f_{\mu_l}(\boldsymbol{h}_{l+1})$ and $f_{\sigma_l}(\boldsymbol{h}_{l+1})$ are applied by NNs with parameters $\theta_l = \{\theta_{\mu_l}, \theta_{\sigma_l}\}$. The result is equivalent as learning the mean and covariance of autoregressive variables (see Figure 4 for illustrative details). However, thanks to this trick, we relax the dependencies among the latent variables, resulting in a smoother joint posterior density $p(\boldsymbol{\epsilon}|\boldsymbol{x}_O, \boldsymbol{y}_O)$. Performing the inference over $\boldsymbol{\epsilon} = \{\boldsymbol{\epsilon}_1, ..., \boldsymbol{\epsilon}_L\}$ leads to a better posed basis for running our HMC optimization, detailed in section 3.5, and avoids the necessity of employing more advanced HMC samplers like [5, 16]. We include further details on the pathological behavior and demonstration of the effectiveness of our solution in Section B.3 of the Supplementary. Provided the promising results we obtain in Section 5, we propose our reparameterization trick as a novel contribution for solving the pathological behavior of HMC combined with hierarchical VAEs.

For the sake of simplicity, we name the generative parameters as $\theta = \{\theta_z, \theta_y, \theta_1, ..., \theta_L\}$. The dependency ELBO under this hierarchical reparameterized model becomes

$$\mathcal{L}_{VI}(\boldsymbol{x}_O, \boldsymbol{y}_O; \{\theta, \psi\}) = \mathbb{E}_{q_\psi}\left[\log \frac{p_\theta(\boldsymbol{z}_O, \boldsymbol{y}_O, \boldsymbol{\epsilon})}{q_\psi(\boldsymbol{\epsilon}|\boldsymbol{z}_O, \boldsymbol{x}_O, \boldsymbol{y}_O)}\right] =$$

$$\mathbb{E}_{q_\psi}\left[\log p_\theta(\boldsymbol{z}_O|\boldsymbol{h}_1) + \log p_\theta(\boldsymbol{y}_O|\hat{\boldsymbol{x}}, \boldsymbol{h}_1)\right] - \sum_{l=1}^{L} D_{\mathrm{KL}}\left(q_\psi(\boldsymbol{\epsilon}_l|\boldsymbol{x}_O, \boldsymbol{y}_O)||p(\boldsymbol{\epsilon}_l)\right). \tag{6}$$

We name $\boldsymbol{r}_l$ the hidden representation at each layer, and defining $\boldsymbol{r}_0 = \{\boldsymbol{x}_O, \boldsymbol{y}_O\}$, we employ NNs with parameters $\psi_{r_l}$ for computing $\boldsymbol{r}_l = f_r(\boldsymbol{r}_{l-1})$. These vectors are mapped into the parameters of the variational posterior $q_{\psi_l}(\boldsymbol{\epsilon}_l|\boldsymbol{x}_O, \boldsymbol{y}_O)$, using NNs for computing the mean as $g_{\mu_l}(\boldsymbol{r}_l)$ and the covariance as $g_{\sigma_l}(\boldsymbol{r}_l)$, with parameters $\psi_{\mu_l}$ and $\psi_{\sigma_l}$. With compactness purposes, we will denote the encoder parameters as $\psi = \{\psi_1, ..., \psi_L\}$, with $\psi_l = \{\psi_{r_l}, \psi_{\mu_l}, \psi_{\sigma_l}\}$.

## 3.5 HMC over the hierarchical density

In recent works, HMC has been combined with deep generative models for improving the inference of the latent variables by obtaining better samples from the posterior [7, 19]. In this work, we propose to transcend these previous approaches and build a generalized method for sampling from complicated, hierarchical latent structures composed by several layers. Inspired by [6] and their method for sampling from the posterior within a vanilla VAE framework while tuning the HMC hyperparameters, we follow a procedure for training the dependency model where i) during a pre-training stage, the encoder and decoder are optimized using standard VI and the ELBO from equation (6), and ii) using the pre-trained encoder for starting from a good proposal [19], HMC samples are obtained to follow the true posterior and jointly optimize the generative model and the HMC hyperparameters. In Figure 1 (b) we include an illustrative example.

We denote by $q_\phi^{(T)}(\boldsymbol{\epsilon}|\boldsymbol{z}_O, \boldsymbol{x}_O, \boldsymbol{y}_O)$ the implicit distribution for the posterior after $T$ HMC steps. The hyper-parameters of HMC are named $\phi$, a $T \times d$ matrix containing the step sizes for each dimension at each step of the chain. Within this perspective, the hyper-parameters can be optimized using variational inference by maximizing the ELBO

$$\mathbb{E}_{q_\phi^{(T)}(\boldsymbol{\epsilon})}[\log p(\boldsymbol{z}_O, \boldsymbol{y}_O, \boldsymbol{\epsilon})] + H[q_\phi^{(T)}(\boldsymbol{\epsilon}|\boldsymbol{x}_O, \boldsymbol{y}_O)], \tag{7}$$

where the first part is the HMC objective, and can be easily estimated via Monte Carlo

$$\mathcal{L}_{HMC}(\boldsymbol{z}_O, \boldsymbol{y}_O; \{\theta, \psi, \phi\}) = \mathbb{E}_{q_\phi^{(T)}(\boldsymbol{\epsilon})}[\log p_\theta(\boldsymbol{z}_O|\boldsymbol{h}_1) + \log p_\theta(\boldsymbol{y}_O|\hat{\boldsymbol{x}}, \boldsymbol{h}_1) + \sum_{l=1}^{L} p(\boldsymbol{\epsilon}_l^{(T)})]. \tag{8}$$

Nevertheless, the entropy term $H[q_\phi^{(T)}(\boldsymbol{\epsilon}|\boldsymbol{x}_O, \boldsymbol{y}_O)]$ in Equation (7) is not tractable since we are not able to explicitly evaluate the distribution $q_\phi^{(T)}(\boldsymbol{\epsilon}|\boldsymbol{x}_O, \boldsymbol{y}_O)$. Although optimizing the first term might result in a well-posed algorithm, this would bring consequences that must be considered. Namely, without a proper regularization term, and in case the initial proposal $q_\phi^{(T)}(\boldsymbol{\epsilon}|\boldsymbol{x}_O, \boldsymbol{y}_O)$ is concentrated in high density regions, the chains would barely move from the initial state and only these regions with high density would be explored (see Section B.1 of the Supplementary for illustrative details). To cope with this problem, we define an inflation parameter $\boldsymbol{s}$ to increase the variance of the proposal $q_\phi(\boldsymbol{\epsilon}|\boldsymbol{z}_O, \boldsymbol{x}_O, \boldsymbol{y}_O)$ given by the Gaussian encoder, ending in the proposal $\prod_{l=1}^{L} \mathcal{N}(g_{\mu_l}(\boldsymbol{r}_l), \; s_l \cdot g_{\sigma_l}(\boldsymbol{r}_l))$. Whilst in [6] the authors define this parameter as a scalar factor applied to all the latent dimensions, in our work, we extend this to apply a different inflation factor at each latent level $\boldsymbol{s} = \{s_1, ..., s_L\}$. In order to tune these inflations we ensure a wider coverage of the density by minimizing the Sliced Kernelized Stein Discrepancy (SKSD) [17]

$$\mathcal{L}_{SKSD}(\boldsymbol{x}_O, \boldsymbol{y}_O; \boldsymbol{s}) = \text{SKSD}\left(q_\phi^{(T)}(\boldsymbol{\epsilon}|\boldsymbol{z}_O, \boldsymbol{x}_O, \boldsymbol{y}_O; \boldsymbol{s}), \, p(\boldsymbol{\epsilon}|\boldsymbol{z}_O, \boldsymbol{x}_O, \boldsymbol{y}_O)\right), \tag{9}$$

which fits perfectly our requirements, since only requires samples from HMC and gradients $\nabla_\epsilon \log p(\boldsymbol{z}_O, \boldsymbol{y}_O, \boldsymbol{\epsilon})$ for measuring a discrepancy between the implicit and the true posterior. Further, the SKSD performs better than other discrepancies like [31] in high dimensional latent spaces.

We provide in Section B.1 of the Supplementary a toy demonstration on the efficacy of the HMC optimization, and in Section B.2 a demonstration of the optimization convergence.

## 3.6 HH-VAEM optimization

The optimization of the HH-VAEM is divided into three stages. In a first stage, we train one independent *marginal* VAE per dimension. In a second stage, the *dependency* VAE is trained, using as inputs the concatenation of the encoded dimensions $\boldsymbol{z} = \{z_1, ..., z_D\}$ and the target $\boldsymbol{y}$. Finally, in a third stage, the HMC hyperparameters, the decoder and the predictor are tuned using the HMC objective, the inflation parameter is trained using the SKSD discrepancy, and the encoder is trained used the ELBO. The pseudocode for HH-VAEM training is shown in Algorithm 1.

## 3.7 Computational cost

In a VAE, the data is fed to the encoder with aim at obtaining the variational parameters for sampling from the Gaussian approximated posterior. In our method, the data is similarly encoded to obtain the

initial Gaussian proposal $q^{(0)}$, and the samples from this distribution are updated for $T$ cycles to obtain the implicit $q^{(T)}$ using HMC. Within each of these iterations, $L$ leapfrog steps (1) are executed. For each of these steps, the computation of the gradients $\nabla_{\epsilon_l} \log p(z, y, \epsilon_l)$ and $\nabla_{\epsilon_{l+1}} \log p(z, y, \epsilon_{l+1})$ is required. To obtain these gradients, we need to i) compute the parameters of the likelihood $p(z, y|\epsilon)$ that are given by the decoder $(p(z|h_1))$ and predictor $(p(y|\hat{x}, h_1))$, ii) evaluate the likelihood and iii) perform the automatic differentiation. Thus, for running our method, an additional cost from both decoding and performing differentiation a total of $2TL$ times is introduced.

By jointly optimizing the HMC hyperparameters we are able to achieve faster convergence with smaller lengths. To reduce the computational cost, we optimize the hyperparameters in a final training stage, since convergence is rapidly achieved (as demonstrated empirically in Section B.2 of the Supplementary). At test, samples from the Gaussian $q^{(0)}$ (faster and cheaper), or from HMC $q^{(T)}$ (slower and better) can be used to fit computational constraints.

## 4   Sampling-based Active Learning

Considering that the input data are tuples of observed and missing features $\{x_O, x_U\}$, our Active Learning framework follows [33, 34] and determines which feature $x_i \in x_U$ will

---

**Algorithm 1** Training algorithm for HH-VAEM

**Input:** data $\left( x_O^{(1:N)}, y_O^{(1:N)} \right)$, steps: $T_d, T_{VI}, T_{HMC}$
**Parameters:** $\gamma, \theta, \psi, \phi, s$
STAGE 1: MARGINAL VAEs
**for** $d = 1$ **to** $D$ **do**
    Initialize marginal VAE $\{\theta_d, \gamma_d\}$
    **for** $t = 1$ **to** $T_d$ **do**
        $\gamma_d^{t+1}, \theta_d^{t+1} \leftarrow \text{Adam}_{\gamma_d^t, \theta_d^t}(\mathcal{L}_d)$
    **end for**
**end for**
STAGE 2: DEPENDENCY VAE
**for** $t = 1$ **to** $T_{VAE}$ **do**
    $\theta^{t+1}, \psi^{t+1} \leftarrow \text{Adam}_{\theta^t, \psi^t}(\mathcal{L}_{VI})$
**end for**
STAGE 3: JOINTLY OPTIMIZING VAE + HMC
**for** $t = 1$ **to** $T_{HMC}$ **do**
    $\psi^{t+1} \leftarrow \text{Adam}_{\psi^t}(\mathcal{L}_{VI})$
    $\theta^{t+1}, \phi^{t+1} \leftarrow \text{Adam}_{\theta^t, \phi^t}(\mathcal{L}_{HMC})$
    $s^{t+1} \leftarrow \text{Adam}_{s^t}(\mathcal{L}_{SKSD})$
**end for**

---

enhance the prediction of the target $y$ the most for a particular $x_O$. Concretely, in a Sequential Active Information Acquisition (SAIA) scenario, this decision is taken sequentially to optimally increase knowledge and accurately predict $y$. From a information theoretical perspective, this task can be performed recurrently by maximizing a reward function $R$ at each step $d$. This reward might represent abstract quantities of interest like the cost or benefit of acquiring $x_i$ (depending on the sign). In Bayesian experimental analysis, $R$ is the expected gain of information [29]. Following [3], we can define it as

$$R(i, x_O) = \mathbb{E}_{p(x_i|x_O)} D_{\text{KL}} \left( p(y|x_i, x_O) p(y|x_O) \right), \tag{10}$$

where $i$ is the index of each unobserved feature. Intuitively, this quantity can be interpreted as the expected change in the predictive distribution when $x_i$ is observed. The reward needs to be estimated via Monte Carlo by sampling from $p(x_i|x_O)$. With a robust generative model that handles missing data like HH-VAEM, these samples are easily obtained: first, using HMC, we sample $\epsilon^{(T)}$ from $p(\epsilon|x_O)$. Second, we decode these samples to obtain $x_i$ from $p(z|h_1)$ and $p(x_i|z_d)$. Nonetheless, the reward defined in (10) is intractable since both $p(y|x_i)$ and $p(y|x_i, x_O)$ are intractable: computing them requires to integrate out the latent variables. This motivates the authors of [33, 34] to present a transformation of the reward for being computed in the latent space using the encoder network. Although they prove that this transformation effectively provides a good estimation in several datasets, we demonstrate that for low dimensional targets (commonly one or two dimensions), an approximation using histograms is more effective. Concretely, the reward in (10) can be rewritten as

$$D_{\text{KL}} \left[ p(y, x_i|x_O) || p(y|x_O) p(x_i|x_O) \right] = \mathcal{I}(y; x_i | x_O), \tag{11}$$

While a set of advanced non-parametric estimators of the mutual information are available [26, 42], many are not easily adapted for parallelization. We demonstrate that the simplest one,

$$\hat{I}(y; x_i | x_O) \approx \sum_{ij} p(i, j) \log \frac{p(i, j)}{p_x(i) p_y(j)}, \tag{12}$$

based on binning the $x_i$ and $y$ domains, is effective and easy to parallelize. In the equation, $p_x(i) = n_x(i)/N$, $p_y(j) = n_y(j)/N$ and $p(i, j) = n(i, j)/N$ are the relative frequencies that approximate

|  | Bank | Insurance | Avocado | Naval | Yatch | Diabetes | Concrete | Wine | Energy | Boston |
|---|---|---|---|---|---|---|---|---|---|---|
| VAEM | $2.84 \pm 0.07$ | $1.81 \pm 0.03$ | $1.89 \pm 0.01$ | $0.55 \pm 0.05$ | $3.15 \pm 0.28$ | $2.78 \pm 0.16$ | $2.45 \pm 0.26$ | $3.01 \pm 0.61$ | $2.09 \pm 0.10$ | $2.01 \pm 0.23$ |
| MIWAEM | $2.74 \pm 0.05$ | $1.88 \pm 0.04$ | $1.92 \pm 0.04$ | $0.57 \pm 0.03$ | $2.66 \pm 0.11$ | $2.55 \pm 0.09$ | $2.34 \pm 0.51$ | $2.76 \pm 0.48$ | $2.06 \pm 0.14$ | $1.94 \pm 0.23$ |
| H-VAEM | $2.82 \pm 0.06$ | $1.80 \pm 0.04$ | $1.89 \pm 0.01$ | $0.48 \pm 0.06$ | $3.06 \pm 0.31$ | $2.74 \pm 0.09$ | $2.42 \pm 0.21$ | $2.85 \pm 0.56$ | $1.72 \pm 0.11$ | $1.89 \pm 0.24$ |
| HMC-VAEM | $2.69 \pm 0.05$ | $1.77 \pm 0.06$ | $1.89 \pm 0.02$ | $0.49 \pm 0.07$ | $\mathbf{2.21 \pm 0.24}$ | $2.72 \pm 0.20$ | $2.28 \pm 0.29$ | $2.83 \pm 0.46$ | $1.73 \pm 0.05$ | $1.83 \pm 0.16$ |
| **HH-VAEM** | $\mathbf{2.63 \pm 0.04}$ | $\mathbf{1.75 \pm 0.03}$ | $\mathbf{1.88 \pm 0.05}$ | $\mathbf{0.40 \pm 0.05}$ | $2.47 \pm 0.27$ | $\mathbf{2.54 \pm 0.13}$ | $\mathbf{2.28 \pm 0.09}$ | $\mathbf{1.90 \pm 0.17}$ | $\mathbf{1.71 \pm 0.04}$ | $\mathbf{1.83 \pm 0.11}$ |

Table 1: Test NLL of the unobserved features for our model and baselines.

|  | Bank | Insurance | Avocado | Naval | Yatch | Diabetes | Concrete | Wine | Energy | Boston |
|---|---|---|---|---|---|---|---|---|---|---|
| VAEM | $0.56 \pm 0.06$ | $1.20 \pm 0.03$ | $1.18 \pm 0.02$ | $2.69 \pm 0.01$ | $0.61 \pm 0.02$ | $1.59 \pm 0.19$ | $1.07 \pm 0.09$ | $0.28 \pm 0.09$ | $0.61 \pm 0.14$ | $0.85 \pm 0.21$ |
| MIWAEM | $0.51 \pm 0.03$ | $1.15 \pm 0.03$ | $1.15 \pm 0.03$ | $2.70 \pm 0.01$ | $0.60 \pm 0.03$ | $\mathbf{1.36 \pm 0.10}$ | $0.95 \pm 0.22$ | $0.28 \pm 0.13$ | $0.54 \pm 0.12$ | $0.80 \pm 0.21$ |
| H-VAEM | $0.50 \pm 0.03$ | $1.06 \pm 0.02$ | $1.18 \pm 0.02$ | $2.68 \pm 0.01$ | $0.60 \pm 0.02$ | $1.71 \pm 0.14$ | $1.02 \pm 0.09$ | $0.26 \pm 0.11$ | $0.46 \pm 0.14$ | $0.90 \pm 0.22$ |
| HMC-VAEM | $0.52 \pm 0.02$ | $1.00 \pm 0.03$ | $1.12 \pm 0.03$ | $2.71 \pm 0.01$ | $\mathbf{0.52 \pm 0.15}$ | $1.55 \pm 0.29$ | $0.95 \pm 0.26$ | $0.28 \pm 0.09$ | $0.41 \pm 0.07$ | $0.71 \pm 0.13$ |
| **HH-VAEM** | $\mathbf{0.49 \pm 0.03}$ | $\mathbf{0.93 \pm 0.06}$ | $\mathbf{1.10 \pm 0.01}$ | $\mathbf{2.62 \pm 0.01}$ | $0.56 \pm 0.02$ | $1.38 \pm 0.18$ | $\mathbf{0.95 \pm 0.08}$ | $\mathbf{0.20 \pm 0.04}$ | $\mathbf{0.32 \pm 0.05}$ | $\mathbf{0.55 \pm 0.04}$ |

Table 2: Test NLL of the predicted target for our model and baselines.

$p(x)$, $p(y)$ and $p(x,y)$ for each bin. Thus, $n_x(i)$, $n_y(j)$ and $n(i,j)$ are the number of samples inside each interval. The number of bins defines the width of uniformly distributed intervals over $x_d$ and $\boldsymbol{y}$ supports. Since this estimator is sampling-based, under certain conditions (namely, if all densities exist as proper functions), Equation (13) indeed converges to $\mathcal{I}(\boldsymbol{y}; x_i \,|\, \boldsymbol{x}_O)$ if we first let the number of samples $N \to \infty$ [26].

# 5   Experiments

The evaluation of the HH-VAEM model is organized into three quantitative experiments and one qualitative experiment. The ablation study includes the validation of our proposed HMC-based, hierarchical model with respect to the Gaussian and one-layered alternatives. Namely, the comparison is performed with the following baseline models:

- *VAEM*: 1 layer, Gaussian-based VAEM [33] (without including the Partial VAE).
- *MIWAEM*: 1 layer, Gaussian-based, importance weighted IWAEM (VAEM + IWAE [36]).
- *H-VAEM*: 2 layers, Gaussian-based VAEM.
- *HMC-VAEM*: 1 layer, HMC-based (with our optimization method) VAEM.

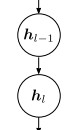

(a) Autoregressive

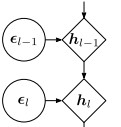

(b) Reparameterization

Figure 4: AR (a) vs reparameterized (b).

|  | VAE | MIWAE | H-VAE | HMC-VAE | **HH-VAE** |
|---|---|---|---|---|---|
| MNIST | $0.124 \pm 0.001$ | $0.121 \pm 0.001$ | $0.119 \pm 0.001$ | $0.101 \pm 0.004$ | $\mathbf{0.094 \pm 0.003}$ |
| F-MNIST | $0.162 \pm 0.002$ | $0.160 \pm 0.002$ | $0.156 \pm 0.002$ | $0.150 \pm 0.002$ | $\mathbf{0.144 \pm 0.002}$ |

Table 3: Test NLL of the unobserved features for the MNIST datasets.

|  | VAE | MIWAE | H-VAE | HMC-VAE | **HH-VAE** |
|---|---|---|---|---|---|
| MNIST | $0.153 \pm 0.009$ | $0.151 \pm 0.007$ | $0.146 \pm 0.006$ | $0.067 \pm 0.007$ | $\mathbf{0.056 \pm 0.019}$ |
| F-MNIST | $0.501 \pm 0.012$ | $0.496 \pm 0.008$ | $0.494 \pm 0.007$ | $0.357 \pm 0.060$ | $\mathbf{0.337 \pm 0.069}$ |

Table 4: Test NLL of the predicted target for the MNIST datasets.

|  | VAE | MIWAE | H-VAE | HMC-VAE | **HH-VAE** |
|---|---|---|---|---|---|
| MNIST | $0.953 \pm 0.004$ | $0.953 \pm 0.003$ | $0.953 \pm 0.003$ | $0.978 \pm 0.003$ | $\mathbf{0.981 \pm 0.005}$ |
| F-MNIST | $0.824 \pm 0.005$ | $0.824 \pm 0.004$ | $0.824 \pm 0.004$ | $0.869 \pm 0.015$ | $\mathbf{0.876 \pm 0.017}$ |

Table 5: Test accuracy of the predicted digits for the MNIST datasets.

For all the models, we manually introduce missing data in the training set by randomly setting per data point a feature as missing with a probability sampled uniformly in the interval $[0.01, 0.99]$ within each batch. Both the input data $\boldsymbol{x}$ and the target $\boldsymbol{y}$ can be missing. For the test set, a fixed probability of 0.5 leads to about half of the input data being observed, whilst the target is completely unobserved.

For the quantitative experiments, a total of 10 UCI datasets [11] that include mixed-type data are employed for the evaluation. We include both MNIST [27] and Fashion-MNIST [54] datasets for evaluating our model in higher dimensional observational and latent spaces and bigger architectures (3 layered convolutional nets for encoder/decoder). For the qualitative results, we evaluate our model in the image inpainting task on MNIST and CelebA [32]. For the three image datasets, the marginal

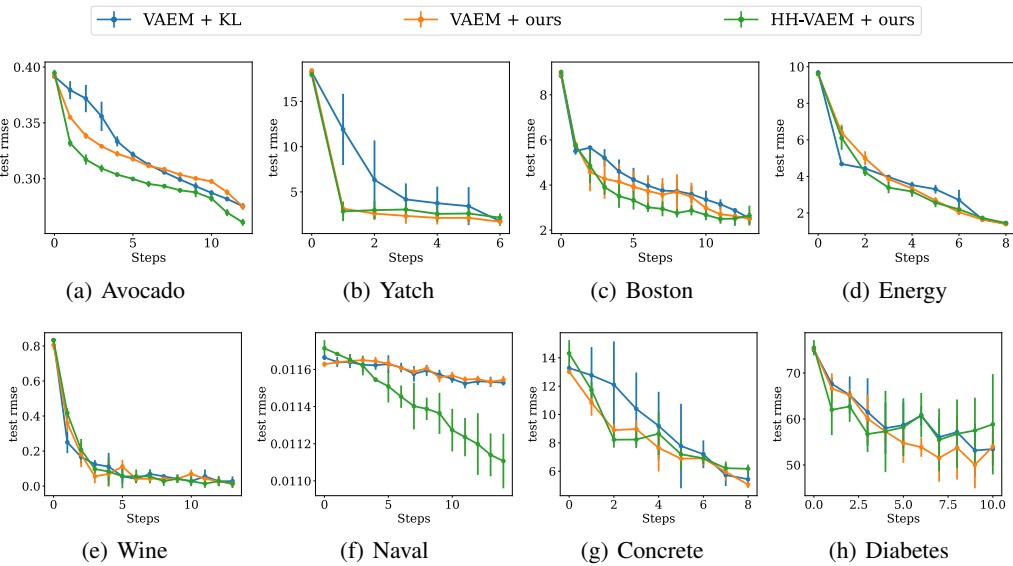


(a) Avocado     (b) Yatch     (c) Boston     (d) Energy



(e) Wine     (f) Naval     (g) Concrete     (h) Diabetes


Figure 5: SAIA curves. Horizontal axis shows number of discovered features. Vertical axis is RMSE.

VAEs are not included and the dependency VAE is fed directly with the Bernoulli-distributed pixels. We name this model HH-VAE, and similarly, the baselines are renamed as VAE, MIWAE, HMC-VAE and H-VAE. Extended experiments and validations are provided in the Supplementary. The source code for reproducing our work is available at `https://github.com/ipeis/HH-VAEM`.

## 5.1 Mixed type conditional data imputation

In order to evaluate the performance of the model in terms of data imputation, we opt by computing the negative log likelihood of the unobserved features. We make use of the Monte Carlo approximation

$$\log p(\boldsymbol{x}_U|\boldsymbol{x}_O) \approx \log \mathbb{E}_{\boldsymbol{\epsilon} \sim q^{(T)}(\boldsymbol{\epsilon}|\boldsymbol{x}_O)} \left[ p(\boldsymbol{x}_U|\boldsymbol{\epsilon}) \right] \approx \log \frac{1}{k} \sum_i^k p(\boldsymbol{x}_U|\boldsymbol{\epsilon}_i), \tag{13}$$

which is averaged over features in order to compare the imputation performance with the baselines. Additionally, we include in Section C.3 similar results averaging each of the considered likelihoods. Results on the 10 UCI datasets and the MNIST datasets are included in Tables 1 and 3, showing that for most of the datasets, incremental improvement is obtained: VAEM < H-VAEM < HMC-VAEM < HH-VAEM. Extended results with the imputation error, included in Section C.1, corroborate this.

## 5.2 Target prediction

For this experiment, we compute the negative log likelihood of the target under the predictive distribution using the same procedure as in the previous section. Results included in Tables 2 and 4 show the same incremental improvement in the prediction task.

## 5.3 Sequential active information acquisition (SAIA)

In this experiment, our HH-VAEM model and our acquisition method are evaluated in a SAIA task. Starting by predicting from completely unobserved inputs, at each step, the missing feature that maximizes the reward is acquired. Figure 5 shows the error curves for the UCI datasets. Blue lines correspond to the Gaussian-based reward proposed by [33]. Orange lines are our sampling-based reward in a VAEM framework. Green lines correspond to HH-VAEM with our reward. In most of the cases, our model and acquisition method obtains lower errors and faster discovery of information.

## 5.4 Conditional image inpainting

We include in this experiment qualitative results when comparing our model with the baselines on the image conditional inpainting task. We include results for MNIST and CelebA in Figure 6 (a)

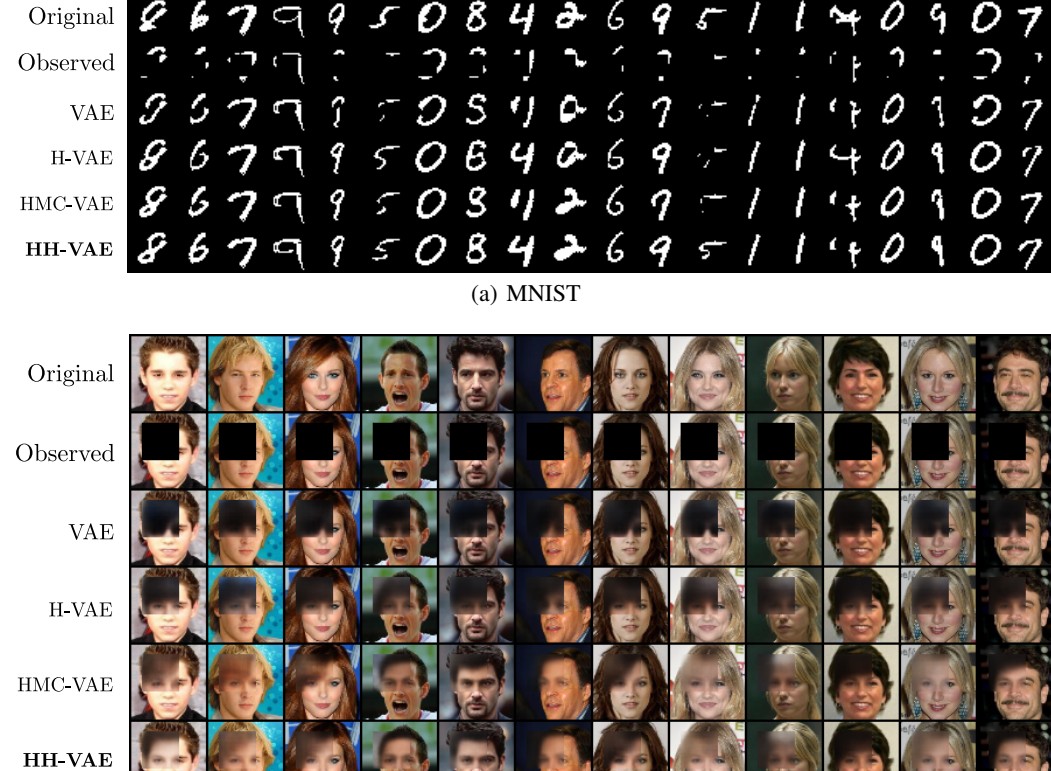

(a) MNIST

(b) CelebA

Figure 6: Image conditional inpainting on MNIST (a) and CelebA (b). First row: original images from the test set. Second row: input to the model, $x_O$, with a black square missing mask $x_U$ manually introduced. Third, fourth, fifth and sixth rows: imputed $\hat{x}_U$ for VAE, H-VAE, HMC-VAE and HH-VAE, our model, where $\hat{x}_U$ is decoded from samples of the approximate posterior (Gaussian for VAE and H-VAE, or HMC-based for HMC-VAE and HH-VAE).

and (b), respectively, that show the superiority of our method. First, the hierarchical Gaussian model (fourth row) considerably improves the one-layered Gaussian alternative. Second, the HMC-based methods (two last rows) vastly improve the Gaussian methods. Third, in some specific cases, an extra improvement is added by HH-VAE with respect to the one-layered HMC-based model (columns 2, 4, 8, 9, 11, 12 and 13 in Figure 6 (a) or columns 1, 2, 5 and 12 in (b)).

# 6   Conclusion

We presented HH-VAEM, to our knowledge, the first hierarchical VAE for mixed-type incomplete data that uses HMC with automatic hyper-parameter tuning for improved inference. We provide both quantitative and qualitative experiments that demonstrate its superiority with respect the baselines in the tasks of missing data imputation and supervised learning, placing HH-VAEM as a robust model for real-world datasets. Further, we have developed a novel sampling-based technique for dynamic feature selection that outperforms the Gaussian-based alternatives and results in an efficient method for active learning in deep generative models.

## Acknowledgments and Disclosure of Funding

Ignacio Peis acknowledges support from Spanish government Ministerio de Ciencia, Innovación y Universidades under grants FPU18/00516, RTI2018-099655-B-I00, EST21/00467 and PID2021-123182OB-I00 and from Comunidad de Madrid under grant Y2018/TCS-4705 PRACTICO-CM. José Miguel Hernández-Lobato acknowledges support from Boltzbit.

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
