where $i$ is the index of each unobserved feature. Intuitively, this quantity can be interpreted as the expected change in the predictive distribution when $\boldsymbol{x}_i$ is observed. The reward needs to be estimated via Monte Carlo by sampling from $p(x_i|\boldsymbol{x}_O)$. With a robust generative model that handles missing data like HH-VAEM, these samples are easily obtained: first, using HMC, we sample $\boldsymbol{\epsilon}^{(T)}$ from $p(\boldsymbol{\epsilon}|\boldsymbol{x}_O)$. Second, we decode these samples to obtain $x_i$ from $p(\boldsymbol{z}|\boldsymbol{h}_1)$ and $p(x_i|z_d)$. Nonetheless, the reward defined in (10) is intractable since both $p(\boldsymbol{y}|x_i)$ and $p(\boldsymbol{y}|x_i, \boldsymbol{x}_O)$ are intractable: computing them requires to integrate out the latent variables. This motivates the authors of [33, 34] to present a transformation of the reward for being computed in the latent space using the encoder network. Although they prove that this transformation effectively provides a good estimation in several datasets, we demonstrate that for low dimensional targets (commonly one or two dimensions), an approximation using histograms is more effective. Concretely, the reward in (10) can be rewritten as

$$D_{\text{KL}}\left[p(\boldsymbol{y}, x_i|\boldsymbol{x}_O)||p(\boldsymbol{y}|\boldsymbol{x}_O)p(x_i|\boldsymbol{x}_O)\right] = \

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

# A  Experimental details

## A.1  Experimental setup

The networks for the encoder of the model with the MNIST datasets are 2 layered Deep CNNs with $\{16, 32, 32\}$ output channels, kernel size 5, stride 2 and padding 2. For the experiments with CelebA, we use 5 layered Deep CNNs with $\{32, 32, 64, 64, 512\}$ output channels, kernel size 4, stride 2 and padding 1, followed by batch norm layers. They are followed by MLPs with 512 hidden units for obtaining the variational parameters for each layer. The decoder that obtains $p_\theta(\boldsymbol{x}|\boldsymbol{h}_1)$ is the symmetric CNN. All the NNs employed in the models trained with UCI datasets are one single layer MLPs with 256 hidden units. The noise variance for Gaussian likelihoods is set up to $0.1$.

We employ learning rates of $1 \times 10^{-3}$ for the models with MLP networks and $2 \times 10^{-4}$ for the convolutional models. For the inflation parameter $\boldsymbol{s}$, we increase to $1 \times 10^{-2}$ for a faster convergence. A batch size of $100$ is used for all the models except for Yatch and Wine dataset, where we use $20$ samples per batch. The number of training steps is $2 \times 10^4$ for Boston, Energy, Wine, Yatch, Concrete, Diabetes and Yatch, and $5 \times 10^4$ for Naval, Avocado, Bank, and Insurance. For MNIST and Fashion-MNIST, we have $1 \times 10^5$ training steps. For CelebA, we use $1, 5 \times 10^5$ training steps. For the marginal VAEs stage, we employ $1 \times 10^3$ training steps. The dimension of the latent variables is $[d_1 = 10, d_2 = 5]$ for Boston, Energy, Wine, Naval Avocado, Bank and Insurance $[d_1 = 4, d_2 = 2]$ for Concrete, Yatch and Diabetes, $[d_1 = 20, d_2 = 10]$ for MNIST and Fashion-MNIST, and $[d_1 = 32, d_2 = 16]$ for CelebA.

We use $LF = 5$ Leapfrog steps in all cases, chains of $T = 10$ for Boston, Energy, Wine, Naval Avocado, Bank, Insurance, MNIST, Fashion-MNIST and CelebA, and $T = 5$ for Concrete, Yatch and Diabetes. The SKSD function is estimated using 30 HMC samples.

For the MNIST and CelebA datasets, the use of Nvidia P100 GPU with Pascal architecture sped up the training with the CNN-based architecture. For the UCI datasets, due to the use of small networks, the differences when using CPU or GPU are almost imperceptible.

## A.2  Datasets information

Apart from the MNIST, Fashion-MNIST and CelebA datasets, a total of 10 UCI datasets have been employed in this work including: Bank Marketing, Insurance Company Benchmark, Avocado sales, Naval Propulsion Plants, Yatch Hydrodynamics, Diabetes, Boston Housing, Wines, Energy efficiency and Bank Marketing.

## A.3  Balancing the KLs

Following [53], we define a short initial warming stage (10% of the total training steps) during the optimization where the KLs for the different layers are balanced according to their magnitude and the corresponding latent dimension, preventing the model for falling into posterior collapse by ignoring deepest layers. A factor is applied to each KL, following

$$\gamma_l = \frac{d_l \, \mathbb{E}_{x \sim B} \left[ \mathrm{KL}(q(\boldsymbol{\epsilon}_l|\boldsymbol{x}) || p(\boldsymbol{\epsilon})) \right]}{\sum_{i=1}^{L} d_i \, \mathbb{E}_{x \sim B} \left[ \mathrm{KL}(q(\boldsymbol{\epsilon}_i|\boldsymbol{x}) || p(\boldsymbol{\epsilon})) \right]}. \tag{14}$$

The factors penalises the fact that a layer might be ignored by making the KL smaller when its magnitude is small compared to the rest layers.

# B  Hamiltonian Monte Carlo with automatic optimization

## B.1  Efficacy of training HMC

We include in this experiment results that demonstrate the efficacy of training the HMC hyperparameters using our proposed gradient-based strategy on 2D densities. In Figure 7, each row correspond to a different example: first row is a *wave* density, second row is a *dual moon* density. In the first column, the initial set up is showed, including the density contour (dark blue), the initial Gaussian proposal contour (light blue) and samples from HMC (green). In both cases, due to the tightness of the proposal, chains do not properly explore the density and get stuck close the initial state. In the

second column, results after training HMC hyperparameters and the inflation parameter are included. Again, in both cases, and more specially in the first one, the inflation of the horizontal variance of the proposal successfully increases to better cover the surface, and final HMC samples vastly improve the exploration. In the third column, the approximations of the HMC objective, the SKSD discrepancy and the inflation parameters over the optimization steps are included.

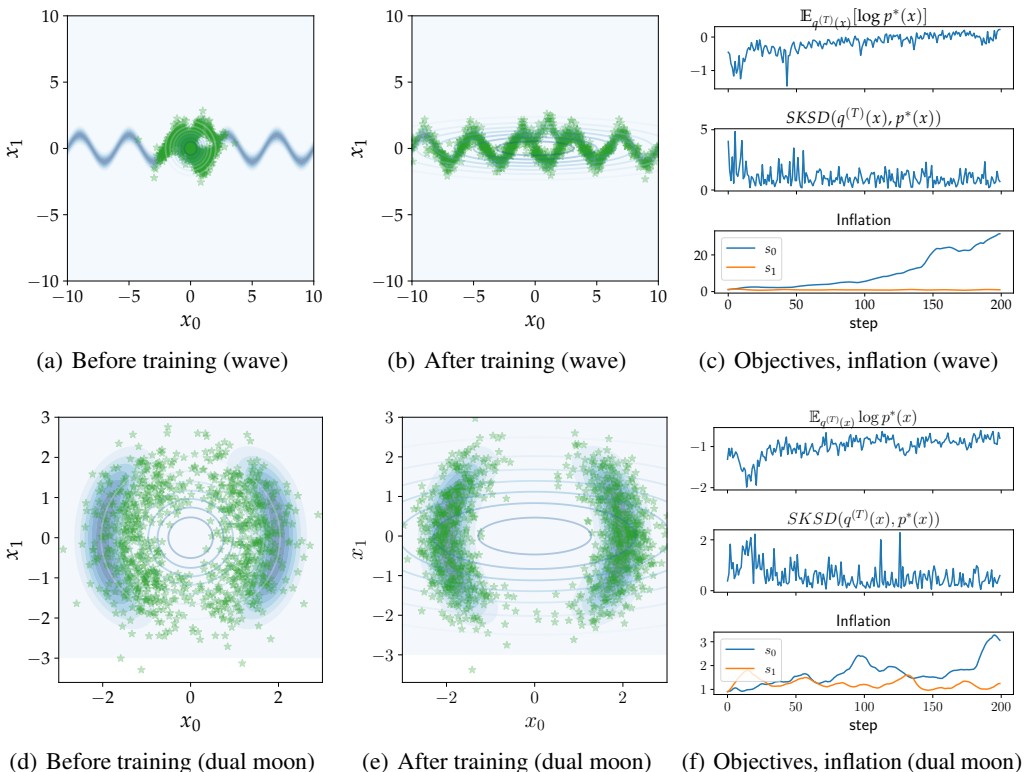

(a) Before training (wave)    (b) After training (wave)    (c) Objectives, inflation (wave)

(d) Before training (dual moon)    (e) After training (dual moon)    (f) Objectives, inflation (dual moon)

Figure 7: Toy example showing the efficacy of training HMC hyperpameters using our method on two densities: wave (top row) with $T = 5$ and dual moon (bottom row) with $T = 10$. Left column illustrates the non desired behavior when chains hardly explore the density and stuck in a small region close to the mass of the tight Gaussian initial proposal (light blue contour ellipses). Right column shows how optimizing the step sizes and the inflation parameter leads to a vast improvement of the exploration. More specifically, (b) justifies scaling each dimension of the target, since the inflation is bigger on the horizontal axis.

## B.2   HMC optimization

We face the computational cost of running HMC by defining a small percentage of training steps for the last stage in Algorithm 1. A 10% of the total training steps for $T_{HMC}$ is sufficient for obtaining the convergence. In Figure 8 we include the validation metrics obtained during the optimization of Yatch dataset, where $T_{HMC} = 2 \times 10^3$.

In figure 9 (a), the mean acceptance rate of the HMC sampler over the training steps and the step sizes (b) are included. The steps are initialized from $U(0.05, 0.2)$. After $2 \times 10^3$ steps, the mean acceptance rate converges to a value closer to $\bar{p}_a = 0.65$, which is defined as the optimal desired acceptance probability [39]. This empirical result provides evidence that, apart from reducing the computational cost, reducing the HMC training step to this value is sufficient for achieving convergence.

## B.3   Reparameterization trick for solving ill-posed HMC with hierarchical densities

We provide in this section strong empirical demonstration on the efficacy of our proposed reparameterization trick. As stated in Section 3.4, naïve implementations of HMC are ill-posed when combined

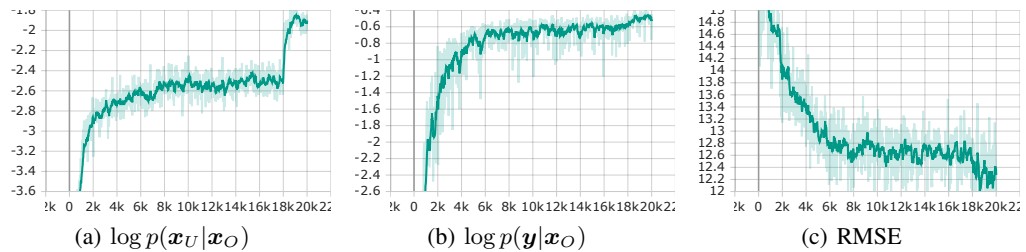

(a) $\log p(\boldsymbol{x}_U|\boldsymbol{x}_O)$        (b) $\log p(\boldsymbol{y}|\boldsymbol{x}_O)$        (c) RMSE

Figure 8: Validation curves during optimization for Naval dataset.

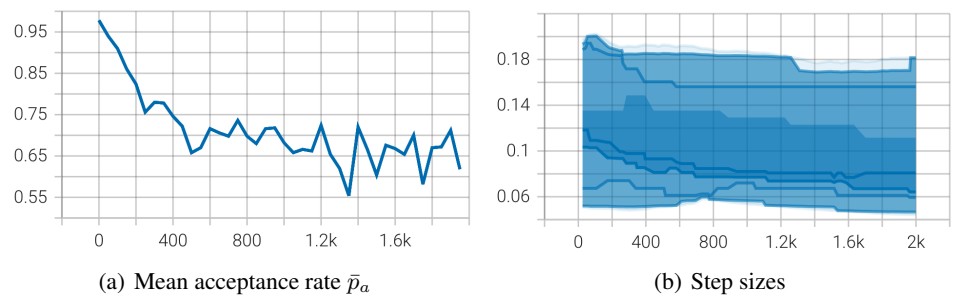

(a) Mean acceptance rate $\bar{p}_a$          (b) Step sizes

Figure 9: Evolution of mean acceptance rate $\bar{p}_a$ (a) and step sizes (b) over the training steps.

with hierarchical densities. The autoregressive correlations lead to non-smooth densities with huge peaks. The large gradients $\nabla_{\boldsymbol{h}_{1:L}} \log p^*(\boldsymbol{h}_{1:L})$ evaluated on these regions inside the Leapfrog steps of Equations 1 make huge modifications of the proposed states. If we denote these diverged states by $\boldsymbol{h}_{1:L}^{(ill)}$, evaluating the objective $\log p^*(\boldsymbol{h}_{1:L}^{(ill)})$ lead to overflow issues. This undesired behavior is what we call *divergence of the Leapfrog integrator*, and is also illustrated by [4] in Figure 35.

In order to avoid the aforementioned overflow issues, we can directly reject these problematic states. Nevertheless, by doing this, HMC will not properly explore the density by the time the states fall into the problematic regions, leading to extremely low acceptance rates. To demonstrate this, we include in Figure 10 (a) with blue line the evolution of the acceptance rates when optimizing HMC with the HH-VAEM variant without reparameterization (as illustrated in Figure 4 (a)). The acceptance rate is extremely low as expected, which is a clear indicator of a poorly mixing sampler.

On the contrary, by using our proposed reparameterization trick (Figure 4 (b)), we are able to make HMC work properly and tune the step sizes, getting closer to the ideal acceptance rate $\bar{p}_a = 0.65$ [39]. Evaluations are applied to the same validation split of the Boston dataset. The step sizes of HMC are initialized from $U(0.05, 0.2)$ in both cases.

Further, we also include in Figure 10 (b) the evolution of the imputation log likelihood metric of Equation (13) during the whole optimization for both approaches. First $1, 8 \times 10^4$ steps correspond to the pretraining stage using only the ELBO. When introducing HMC without the reparameterization, due to the low acceptance rate, the states hardly move from the initial proposal, or move away from the density, and the joint optimization fail. We demonstrate again the effectiveness of the reparameterization trick by observing an increase in this metric.

## C   Extended experiments

### C.1   Deterministic imputation metrics

We include in Table 6 results on the RMSE obtained with other discriminative validated predictors, using mean imputation under the same missing rates. Additionally, we include here the missForest in the baselines, a wide-spread method for missing data imputation using a Random Forest approach [49]. For classification tasks, the error rate is considered. In almost all cases, HH-VAEM outperforms the baselines.

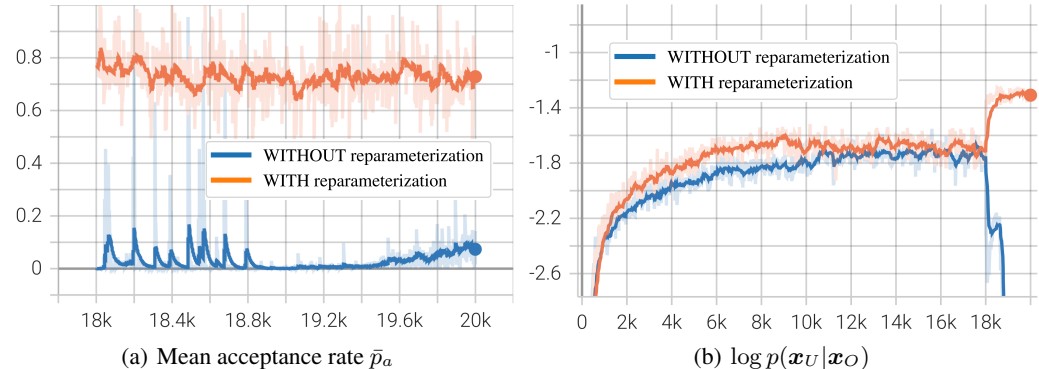

(a) Mean acceptance rate $\bar{p}_a$        (b) $\log p(\boldsymbol{x}_U|\boldsymbol{x}_O)$

Figure 10: Demonstration of the efficacy of our reparameterization trick. Our model is showed with orange lines, and the same model without the reparameterization with blue lines. In (a), the mean acceptance rate of the HMC proposals over the optimization steps is included. We demonstrate that without the reparameterization, HMC is ill-posed and by rejecting the proposals that make the integrator diverge, the acceptance rate is extremely low, thus not properly exploring the density. In (b), the imputation log likelihood metric is showed for the whole optimization. With the reparameterization trick, we successfully solve the pathological issues, leading to a considerable increase of the metric.

| | Bank | Insurance | Avocado | Naval | Yatch | Diabetes | Concrete | Wine | Energy | Boston |
|---|---|---|---|---|---|---|---|---|---|---|
| missForest | $0.64 \pm 0.01$ | $0.61 \pm 0.06$ | $0.59 \pm 0.02$ | $0.30 \pm 0.01$ | $\mathbf{0.86 \pm 0.12}$ | $0.76 \pm 0.08$ | $0.76 \pm 0.07$ | $0.77 \pm 0.11$ | $0.64 \pm 0.08$ | $0.59 \pm 0.06$ |
| VAEM | $0.57 \pm 0.01$ | $0.40 \pm 0.01$ | $0.59 \pm 0.00$ | $0.33 \pm 0.01$ | $0.93 \pm 0.04$ | $0.79 \pm 0.02$ | $0.74 \pm 0.04$ | $0.64 \pm 0.03$ | $0.75 \pm 0.02$ | $0.65 \pm 0.02$ |
| MIWAEM | $0.56 \pm 0.00$ | $0.39 \pm 0.00$ | $0.59 \pm 0.01$ | $0.34 \pm 0.01$ | $0.94 \pm 0.04$ | $0.75 \pm 0.01$ | $0.71 \pm 0.03$ | $0.63 \pm 0.02$ | $0.75 \pm 0.02$ | $0.63 \pm 0.01$ |
| H-VAEM | $0.56 \pm 0.01$ | $0.39 \pm 0.00$ | $0.58 \pm 0.01$ | $0.32 \pm 0.01$ | $0.92 \pm 0.05$ | $0.77 \pm 0.01$ | $0.71 \pm 0.02$ | $0.60 \pm 0.04$ | $0.55 \pm 0.03$ | $0.59 \pm 0.01$ |
| HMC-VAE | $0.55 \pm 0.01$ | $\mathbf{0.38 \pm 0.00}$ | $0.58 \pm 0.01$ | $0.30 \pm 0.02$ | $0.91 \pm 0.05$ | $0.76 \pm 0.03$ | $0.71 \pm 0.02$ | $0.60 \pm 0.02$ | $0.55 \pm 0.02$ | $0.57 \pm 0.02$ |
| HH-VAEM | $\mathbf{0.54 \pm 0.01}$ | $\mathbf{0.38 \pm 0.00}$ | $\mathbf{0.57 \pm 0.00}$ | $\mathbf{0.29 \pm 0.02}$ | $0.90 \pm 0.00$ | $\mathbf{0.75 \pm 0.01}$ | $\mathbf{0.70 \pm 0.01}$ | $\mathbf{0.59 \pm 0.04}$ | $\mathbf{0.54 \pm 0.01}$ | $\mathbf{0.56 \pm 0.03}$ |

Table 6: Test RMSE of the unobserved features for our model and baselines.

## C.2 Likelihood of the observed features

We include in this section results on the negative log likelihood of the observed features

$$\log p(\boldsymbol{x}_O) \approx \log \mathbb{E}_{\boldsymbol{\epsilon} \sim q^{(T)}(\boldsymbol{\epsilon}|\boldsymbol{x}_O)} \left[ p(\boldsymbol{x}_O|\boldsymbol{\epsilon}) \right] \approx \log \frac{1}{k} \sum_i^k p(\boldsymbol{x}_O|\boldsymbol{\epsilon}_i). \tag{15}$$

Results are included in Table 7. In almost all the cases, we confirm the incremental superiority when adding each part of our proposed design.

| | bank | insurance | avocado | naval | yatch | diabetes | concrete | wine | energy | boston |
|---|---|---|---|---|---|---|---|---|---|---|
| VAEM | $0.51 \pm 0.05$ | $0.99 \pm 0.05$ | $0.44 \pm 0.01$ | $0.21 \pm 0.01$ | $0.62 \pm 0.13$ | $0.92 \pm 0.12$ | $0.63 \pm 0.18$ | $0.73 \pm 0.18$ | $1.86 \pm 0.09$ | $0.56 \pm 0.11$ |
| MIWAEM | $0.63 \pm 0.02$ | $1.06 \pm 0.03$ | $0.60 \pm 0.03$ | $0.33 \pm 0.01$ | $0.75 \pm 0.07$ | $1.05 \pm 0.06$ | $0.76 \pm 0.09$ | $0.80 \pm 0.06$ | $1.77 \pm 0.15$ | $0.67 \pm 0.03$ |
| H-VAEM | $0.40 \pm 0.04$ | $0.93 \pm 0.04$ | $0.42 \pm 0.05$ | $0.19 \pm 0.07$ | $0.58 \pm 0.09$ | $0.70 \pm 0.13$ | $0.53 \pm 0.18$ | $0.71 \pm 0.15$ | $0.38 \pm 0.02$ | $0.49 \pm 0.07$ |
| HMC-VAE | $0.37 \pm 0.07$ | $\mathbf{0.92 \pm 0.04}$ | $0.39 \pm 0.06$ | $0.18 \pm 0.05$ | $0.54 \pm 0.10$ | $\mathbf{0.68 \pm 0.07}$ | $0.49 \pm 0.22$ | $\mathbf{0.55 \pm 0.07}$ | $0.40 \pm 0.06$ | $0.41 \pm 0.04$ |
| HH-VAEM | $\mathbf{0.33 \pm 0.03}$ | $0.95 \pm 0.05$ | $\mathbf{0.36 \pm 0.01}$ | $\mathbf{0.17 \pm 0.04}$ | $\mathbf{0.45 \pm 0.04}$ | $0.68 \pm 0.16$ | $\mathbf{0.40 \pm 0.16}$ | $0.64 \pm 0.17$ | $\mathbf{0.37 \pm 0.06}$ | $\mathbf{0.41 \pm 0.04}$ |

Table 7: Test NLL of the observed features for our model and baselines.

## C.3 Heterogeneous likelihoods

In experiment 5.1 we reported an average likelihood across heterogeneous variables. Although this quantity is not a valid joint likelihood probability, we employed this average to provide a fair comparison on models that have been trained on the same heterogeneous likelihoods. In this section, we show the comparison on averaging separately the three considered marginal likelihoods (Gaussian, Bernoulli and Categorical) for two of the biggest datasets considered on Tables 8-10. Again, we show the incremental superiority when adding the different design choices of our model.

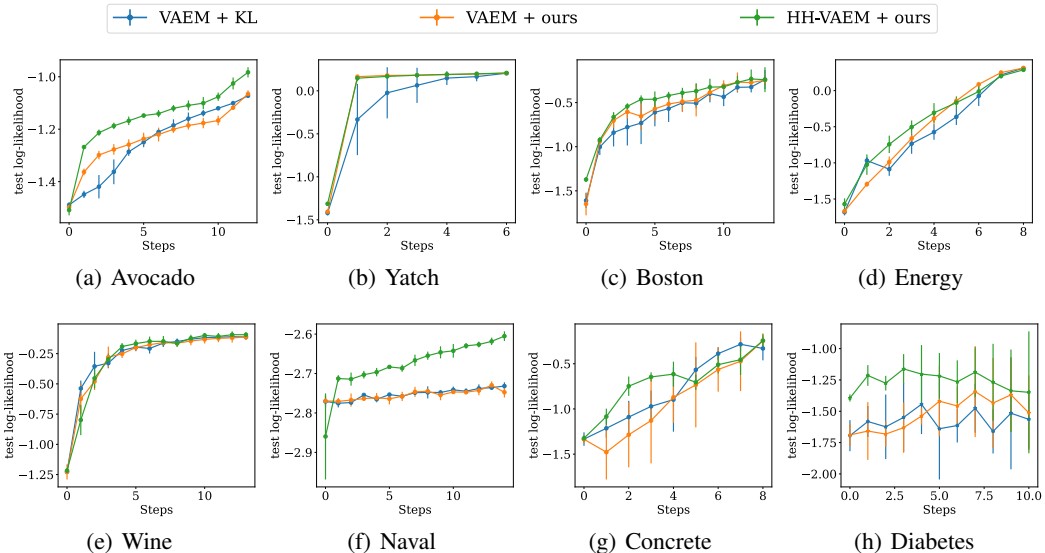

| | VAEM + KL | VAEM + ours | HH-VAEM + ours |

(a) Avocado     (b) Yatch     (c) Boston     (d) Energy

(e) Wine     (f) Naval     (g) Concrete     (h) Diabetes

Figure 11: SAIA log-likelihood curves. Horizontal axis shows acquisition steps (number of discovered features). Vertical axis is the log-likelihood of the target $\log p(\boldsymbol{y}|\boldsymbol{x}_O)$.

| | Bank | Avocado |
|---|---|---|
| VAEM | $0.36 \pm 0.29$ | $0.26 \pm 0.08$ |
| MIWAEM | $0.33 \pm 0.27$ | $0.32 \pm 0.06$ |
| H-VAEM | $0.26 \pm 0.22$ | $0.29 \pm 0.07$ |
| HMC-VAEM | $0.25 \pm 0.21$ | $0.25 \pm 0.08$ |
| **HH-VAEM** | $\mathbf{0.20 \pm 0.22}$ | $\mathbf{0.22 \pm 0.07}$ |

| | Bank | Avocado |
|---|---|---|
| VAEM | $0.13 \pm 0.00$ | $0.06 \pm 0.00$ |
| MIWAEM | $0.15 \pm 0.00$ | $0.09 \pm 0.00$ |
| H-VAEM | $0.11 \pm 0.00$ | $0.07 \pm 0.00$ |
| HMC-VAEM | $0.08 \pm 0.00$ | $0.05 \pm 0.00$ |
| **HH-VAEM** | $\mathbf{0.07 \pm 0.00}$ | $\mathbf{0.04 \pm 0.00}$ |

| | Bank | Avocado |
|---|---|---|
| VAEM | $0.24 \pm 0.16$ | $0.33 \pm 0.00$ |
| MIWAEM | $0.26 \pm 0.17$ | $0.36 \pm 0.00$ |
| H-VAEM | $0.23 \pm 0.16$ | $0.32 \pm 0.00$ |
| HMC-VAEM | $0.22 \pm 0.15$ | $\mathbf{0.30 \pm 0.00}$ |
| **HH-VAEM** | $\mathbf{0.21 \pm 0.15}$ | $\mathbf{0.30 \pm 0.00}$ |

Table 8: Average test Gaussian NLL of the observed features.

Table 9: Average test Bernoulli NLL of the observed features.

Table 10: Average test Cat. NLL of the observed features.

## C.4 SAIA log-likelihoods

In order to extend the results provided in Section 5.3, we include here the log-likelihoods curves when dynamically selecting features using the same procedure (Figure 11).

## C.5 Training times

Table 11 shows the average training time in minutes for each model in the experiments for Tables 1 and 2. The ratio between training times for our method and the Gaussian baselines is approximately between 5 and 10.

| | Bank | Avocado | Yatch | Diabetes | Concrete | Wine | Energy | Boston |
|---|---|---|---|---|---|---|---|---|
| VAEM | $29.92 \pm 0.39$ | $21.49 \pm 1.64$ | $5.89 \pm 0.01$ | $8.79 \pm 0.30$ | $7.70 \pm 0.58$ | $7.92 \pm 0.11$ | $8.18 \pm 0.09$ | $10.01 \pm 0.34$ |
| MIWAEM | $63.33 \pm 6.20$ | $37.17 \pm 2.28$ | $8.21 \pm 0.24$ | $13.29 \pm 0.33$ | $11.51 \pm 0.52$ | $11.81 \pm 0.13$ | $16.71 \pm 0.16$ | $15.01 \pm 0.13$ |
| H-VAEM | $41.44 \pm 0.38$ | $33.83 \pm 0.81$ | $15.84 \pm 0.11$ | $13.38 \pm 0.47$ | $11.80 \pm 0.38$ | $10.61 \pm 0.54$ | $12.71 \pm 1.20$ | $13.74 \pm 1.43$ |
| HMC-VAEM | $281.88 \pm 9.14$ | $356.22 \pm 5.94$ | $23.50 \pm 1.60$ | $50.42 \pm 1.18$ | $42.70 \pm 3.14$ | $63.46 \pm 1.97$ | $90.87 \pm 7.05$ | $103.72 \pm 7.24$ |
| **HH-VAEM** | $316.81 \pm 9.49$ | $388.28 \pm 6.47$ | $27.08 \pm 0.12$ | $68.29 \pm 4.06$ | $65.78 \pm 0.43$ | $79.97 \pm 5.53$ | $140.33 \pm 7.36$ | $129.30 \pm 4.69$ |

Table 11: Training times (in minutes) of our model and baselines.

## C.6 SAIA times

The times for obtaining the SAIA metric curves in Section 5.3 are included in Figure 12. Although the performance is improved with HH-VAEM, it requires considerably longer time than the baselines to evaluate the reward, due to the HMC algorithm for sampling from the better approximated posterior. Future work might be oriented in proposing ways to measure and reduce this gap.

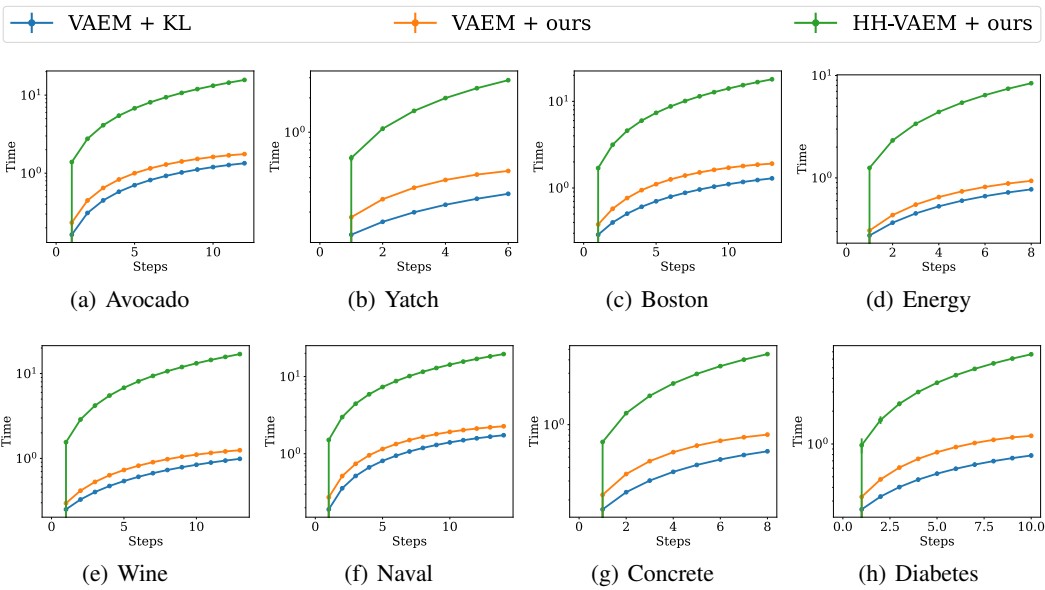

Figure 12: SAIA time curves. Horizontal axis shows acquisition steps (number of discovered features). Vertical axis is the elapsed time.