# OpenReview forum: "Missing Data Imputation and Acquisition with Deep Hierarchical Models and Hamiltonian Monte Carlo"
_NeurIPS.cc/2022/Conference — NeurIPS 2022 Accept_

### Official Review · Reviewer_gfdz · 2022-06-19

**Rating:** 7
**Confidence:** 4
**Soundness:** 4 excellent
**Presentation:** 4 excellent
**Contribution:** 3 good

**Summary:**

After rebuttal from authors:
Thanks for your replies, as well as the new updates/experiments introduced in the new version of the paper.
Considering these, I think this paper is now worth accepting to the conference, and I have updated my score accordingly.

--------------------


This paper introduces the HH-VAEM, a VAE-based model for missing data imputation of heterogeneous data.
The HH-VAEM uses a hierarchical latent space to better model the data, and an HMC sampler to improve approximate inference. The authors show the good performances of the model in several different datasets and tasks.

**Questions:**

See questions in the "Weaknesses" section above.


**Limitations:**

As stated above, there should be more focus on the limitations of the HMC sampler.

**Strengths And Weaknesses:**

STRENGHTS
1. While in many real-world applications data is missing and heterogeneous, much of the literature considers datasets containing fully observed data points of the same type. This paper, on the other hand, provides a novel solution to the problem of imputing and acquiring mixed data.

2.  Experiments show that the model outperforms competing methods on a wide number of datasets.

3. The paper is overall well written.

4. While the presented method is quite complex in its usage of the HMC sampler and the SKSD discrepancy measure, the authors make this model more accessible by focusing on automatic tuning of the step size parameters, and providing code.


WEAKNESSES
1. As opposed to standard VAE training this method is much more complex to understand and implement, and this might limit the impact of this work in the Neurips community.
Given that the authors shared the code this would not be a problem, as long as the method was "plug and play". However, even conidering the effort made by the authors to make it simple to use, I am not convinced it is.
HMC is notoriously hard to use in highly correlated high dimensional spaces, since it is hard to tune the multiple important parameters, and it is also hard to reliably diagnose when the sampler is not mixing properly.
The paper is missing a more detailed discussion on practical considerations on the HMS sampler, for example:
* What are the pitfalls of the HMC sampler? How can one detect a poorly mixing sampler?
* What is the rejection rate of the sampler over the different training epochs? A too low rejection rate might show that the learned step sizes are too small and the sampler is therefore not exploring the space as it should.
* How robust is the method to the choice of the HMC hyperparameters (L, T, initialization for step sizes)

2. Due to the HMC sampler, the training time of the model is much higher than competing methods. The better performances might still justify the overhead in practical applications. However one doubt I have is whether the baselines for non-HMC methods are fair. Have you tried for example using for the standard VAEs models a much higher number of samples to match the computational times? And also for example the IWAE objective from Burda et al, 2015?


Minor comments.
* Typo in line 189: no->not
* line 301 -> what is HH-VAE?

---

> ### Author Response · Authors · 2022-08-02
> **Response to Reviewer gfdz (part 1/2)**
>
>
> We thank you for your valuable suggestions. To address your comments, we have strengthtened the discussion on practical considerations of the HMC sampler in several parts of the paper. Below we specifically answer each of the points:
>
> > On the pitfalls of using the HMC approach and detection of a poorly mixing sampler.
>
> As commented in the paper, the main pitfalls of HMC to be considered in this work are successfully addressed. Namely, as stated in [1, 2], **HMC can be pathological when used for sampling from hierarchical densities**, where huge autoregressive variations increase exponentially with the depth (check simple 2D funnel example on Figure 3 in [2]). For approximating the Hamiltonian dynamics, inside each Leapfrog integrator step, gradients $\nabla_{h_{1:L}} \log p^*(h_{1:L})$ are required. Due to the strong curvature regions, high norm gradients are backpropagated and might eventually explode, ending in overflow issues (Fig. 35 in [1]).
>
> We firstly tried running our HMC method over the hierarchical variables without reparameterization (Fig. 4a). When the states reached the mentioned problematic regions, the integrator diverged and we experienced overflow problems. By rejecting these states, chains got stuck and the hierarchical density was not properly explored (thus, HMC did not improve the Gaussian proposal). **This was a clear indicator of a poorly mixing sampler**.
>
> Instead of dealing with advanced versions of HMC, e.g. Riemann manifold HMC [3] with momentum variance depending on the state position, **we successfully solved this by using our proposed reparameterization design**.  Thanks to this novel design choice, we demonstrate with results that HMC is effective and superior in the considered tasks wrt to the Gaussian alternatives.
>
> > Rejection rate for detection of a poorly mixing sampler
>
> To address your question, **we have included in Section A.4 a new Figure 8 showing the evolution of the acceptance race and the step sizes over the training steps**. As observed in Figure 8, thanks to our hyperparameter tuning, a mean acceptance rate of $\bar{p}_a=0.65$ is achieved at convergence, which is considered as the optimal acceptance rate [1].
>
> > Robustness of the method to the choice of the HMC hyperparameters (L, T, initialization for step sizes)
>
> W.r.t. to the choice of $L$ and $T$, bigger values lead to a more flexible posterior exploration with the disadvantage of an increased computational cost (as explained in Section 3.7). W.r.t. to the initialization for step sizes, we refer to Figure 3 in [5], which is a similar HMC optimization approach. In the mentioned Figure, authors show the KSD between the HMC samples and the target distribution (the wave density illustrated in our experiment, Figure 6 (a) and (b)) before and after
> training the hyperparameters for a large range of initialization step sizes. Authors demonstrate that **the method is robust to the initialization point provided it is not excessively large**, in which case every step in the chain is rejected and there is no gradient signal for learning. We experienced a similar behavior when the initial steps are too big, concretely when sampled from $U(0.2, 0.5)$.

---

> ### Author Response · Authors · 2022-08-02
> **Response to Reviewer gfdz (part 2/2)**
>
>
> > Comparison with a non-HMC method matching the computational times
>
> We compare our method with the MIWAEM alternative, which is how we denote using the IWAE method combined with the VAEM approach. Results on a selection of the datasets are included in the following tables. To match the training times, we choose a number of $S=200$ importance samples. In imputation, the importance weighted approximation is still outperformed by HH-VAEM. In prediction, this occured for half of the datasets considered.
>
>
> **NLL of imputed data**
>
>
> |        | boston           | wine             | concrete         | yatch            |
> |--------|------------------|------------------|------------------|------------------|
> | MIWAEM | $2.20\pm 0.20$  | $3.00 \pm 0.44$ | $2.50 \pm 0.33$ | $2.30 \pm 0.20$ |
> | HHVAEM | $\boldsymbol{1.70 \pm 0.10}$ | $\boldsymbol{1.85 \pm 0.22}$ | $\boldsymbol{2.30 \pm 0.26}$ | $\boldsymbol{2.28 \pm 0.20}$ |
>
> **NLL of predicted data**
> |        | boston           | wine             | concrete           | yatch            |
> |--------|------------------|------------------|--------------------|------------------|
> | MIWAEM | $0.79 \pm 0.13$ | $0.36 \pm 0.06$ | $\boldsymbol{0.87 \pm 0.14}$   | $\boldsymbol{0.50 \pm 0.14}$ |
> | HHVAEM | $\boldsymbol{0.55 \pm 0.04}$ | $\boldsymbol{0.20 \pm 0.04}$ | $0.95 \pm 0.0856$ | $0.56 \pm 0.02$ |
>
>
> > Minor comments
>
> The spelling typo is solved. We name HH-VAE to the version of the model not including the marginal VAEs, thus, not considering mixed-type data. This is the case when evaluating the MNIST datasets, where the dependency VAE is fed directly with the Bernoulli-distributed pixels.
>
> ## References
>
> [1] M. Betancourt. A conceptual introduction to Hamiltonian Monte Carlo. arXiv preprint arXiv:1701.02434, 2017.
>
> [2] M. Betancourt and M. Girolami. Hamiltonian Monte Carlo for hierarchical models. Current
> trends in Bayesian methodology with applications, 79(30):2–4, 2015.
>
> [3] M. Girolami and B. Calderhead. Riemann Manifold Langevin and Hamiltonian Monte Carlo
> methods. Journal of the Royal Statistical Society: Series B (Statistical Methodology), 73(2):123–369
> 214, 2011
>
> [4] R. M. Neal et al. MCMC using Hamiltonian dynamics. Handbook of Markov Chain Monte Carlo,
> 2(11):2, 2011.
>
> [5] A. Campbell, W. Chen, V. Stimper, J. M. Hernandez-Lobato, and Y. Zhang. A gradient
> Based Strategy for Hamiltonian Monte Carlo Hyperparameter optimization. In International
> Conference on Machine Learning, pages 1238–1248. PMLR, 2021.

---

> ### Author Response · Authors · 2022-08-08
> **Rebuttal discussion**
>
> Thanks for the time spent reviewing our paper. We would appreciate if you could let us know if our rebuttal and revised paper addressed your concerns and, if so if you could reconsider your rating for the paper. If you still have concerns, please let us know which ones these are, and we will try to address them again.

---

### Official Review · Reviewer_xZXW · 2022-07-05

**Rating:** 5
**Confidence:** 3
**Soundness:** 3 good
**Presentation:** 2 fair
**Contribution:** 2 fair

**Summary:**

This work introduces HH-VAEM, a method for missing-feature imputation, learning and acquisition with heterogeneous data types.
HH-VAEM is designed to improve the performance of the previous methods on the above tasks by combining the flexibility of hierarchical latent space model and the accuracy of the posterior inference with Hamiltonian MC (HMC).

In particular, HH-VAEM is a variant of VAEM, whose second stage is replaced with a hierarchical generative model trained with HMC decorated with minor tweaks previously proven to be effective.
The originality of the paper is found in i) the combination of VAEM, hierarchical model and HMC, and ii) the specific structure of the hierarchical model: the reparametrized autoregressive latent-space model.

After introducing HH-VAEM, results of numerical experiments are presented to demonstrate quantifiable performance improvement brought by HH-VAEM.

**Questions:**

- What is the fundamental difficulty or non-triviality in the combination of hierarchical model and HMC, which is referred to in the last line of the first page?
- To optimize the HMC hyperparameter, you need to differentiate Eq 9 with respect to $\phi$, which requires higher-order derivatives (up to the L-th) of the generative density function since perturbation on the step size $\phi$ propagates through the entire HMC chain. How do you implement such computation? If there is any approximation, how do you justify it?


**Limitations:**

The limitation is discussed only in terms of computational cost.
I would suggest to add discussion on the extra hyperparameters introduced with the richer hierarchical model in terms of, e.g., the sensitivity.


**Strengths And Weaknesses:**

The strength of this work is that it achieves quantifiable and consistent performance improvements in the empirical experiments.
Specifically, the consistency of the results indicates the non-superficial effectiveness of the combination of the technical elements in HH-VAEM.

The weakness is the insufficiency in discussion and validation of the design choice of HH-VAEM, which reduces the significance of this work.
More specifically, performance improvement is rather trivial if one can properly combine richer model (i.e., hierarchical model) and more accurate inference method (i.e., HMC) provided sufficient amount of data.
Thus, the contribution of this paper should lie in the subtlety of the way they are combined such as the reparametrization trick for the hierarchical model.
Although the authors briefly justify their design choices (e.g., reparametrization in terms of density landscape) upon the introduction of each technical element, none of them is formally validated in the experiment except stating that HH-VAEM is superior than rather random baselines.
More concretely, it would be nice if the baseline methods are organized more carefully for ablation study and the differences in the numbers are discussed in terms of the significance of each of the design choices of HH-VAEM.

Minor issues:
- Experimental detail is insufficient (or difficult to find) to reproduce the results. For example, what is the depth of hierarchical model for HH-VAEM?
- Notational collision: L is used both for the length of HMC and the depth of the hierarchical model.
- I did not get the point of Eq 5 when we have more informative expression Eq 7.
- After the first equality of Eq 7, $y_O$ is missing in the numerator.
- Eq 11 is broken.

---

> ### Author Response · Authors · 2022-08-02
> **Response to Reviewer xZXW (part 1/2)**
>
> We thank you for time considered to provide a high quality review of our paper. We have addressed all your comments and improved our paper in the revised version. Along with our responses, we hope solve all your concerns are solved.
>
> > Information acquisition contribution is not considered
>
> Firstly, we would like to highlight that **one of the principal contributions, our proposed Bayesian information acquisition method, is not commented in this review**. The design of this method is motivated by the improvement of the sampling-based estimator when better quality samples of the posterior of the mode are to be acquired. As we demonstrate in Figure 5, similarly than HH-VAEM does, our novel sampling-based technique benefits from both enriching the prior with the hierarchy and sampling from a vastly better approximation of the posterior (as motivated with Figure 1b).
>
> Below, we address each of your comments:
>
> > Insufficiency in discussion and validation of the design choice.
>
> According to your suggestions, we have decided to include the following changes to better justify, validate and discuss each design choice:
>
> * **We have extended the description of the ablation study and the discussion of the results**.
> * **Section 3.4 has been reorganized for better describing the reparameterization of the latent variables** as a novel contribution for solving the pathological behavior of HMC combined with hierarchical models. We provide details on the non-triviality of this combination by answering your first question. To justify a lack of validation wrt to this design choice, we highlight that **without the proposed reparameterization, practical implementation of HMC is ill-posed**.
> * We enforced the motivation of using HMC for improving all the desired tasks. All these tasks require sampling from the posterior of the model, which is conclusively better approximated by the HMC-based models rather than by Gaussian approximations (Figure 1b). More specifically, our sampling-based proposed method for information acquisition would specially benefit form better quality samples. **We validate the usage of HMC in the ablation study**, wrt to the Gaussian-based version of each model, demonstrating that all the HMC-based models outperform their Gaussian versions.
> * We motivate using of a hierarchical path of latent variables with the improvement of model flexibility that has been vastly demonstrated in previous work. **We validate wrt to one-layered variations for both the Gaussian-based and HMC-based models**. We show that, although it is not as crucial as HMC, in almost all the datasets, the one-layered variations are outperformed by their hierarchical version in the three tasks.
>
>
>
> > Difficulty or non-triviality in the combination of hierarchical model and HMC
>
> **We want to emphasize that combining hierarchical model and HMC is a highly non-trivial tasks, that requires significant work**. As stated in [1, 2], **HMC can be pathological when used for sampling from hierarchical densities**, where huge autoregressive variations increase exponentially with the depth. Inside each Leapfrog integrator step, gradients $\nabla_{h_{1:L}} \log p^*(h_{1:L})$ are required. Due to the strong curvature regions, high norm gradients are backpropagated and might eventually explode, ending in overflow issues (Fig. 35, [1]).
>
> **The naive application of HMC method over the hierarchical variables without reparameterization (Fig. 4a) does not work very well**. When the states reached the mentioned problematic regions, the integrator diverged and we experienced overflow problems. By rejecting these states, chains got stuck and the hierarchical density was not properly explored. Consequently, HMC did not improve the Gaussian proposal. **Given this pathological behavior, we consider this alternative an ill-posed model** and it is not considered for evaluation/validation.
>
> We successfully solved this by using our proposed reparameterization technique. The prior $p(\boldsymbol{\epsilon_1}, ..., \boldsymbol{\epsilon_L}) =  \prod_{l=1}^L p(\boldsymbol{\epsilon_l})$ is no longer autoregressive, thus, the posterior is smoother (do not contains huge variations) and the derivatives are relaxed. Provided the superiority of our proposal demonstrated with the results, **we propose our reparameterization trick as a novel approach for addressing this pathological behavior of HMC in deep hierarchical models**.

---

> ### Author Response · Authors · 2022-08-02
> **Response to Reviewer xZXW (part 2/2)**
>
> > Computation of higher-order derivatives of HMC objective wrt the step sizes $\boldsymbol{\phi}$
>
> Gradients of Eq. (9) wrt $\boldsymbol{\phi}$ can be approximated using Monte Carlo and the reparameterization trick. Nevertheless, due to the discontinuity introduced by the accept/reject step of HMC, a double approximation is required. Ignoring this non-linearity leads to biased gradients. As derived in [4], the term for **the introduced bias is originated from the dependence of the acceptance probability on the hyperparameters, and its expected magnitude is analytically proved to be small**. Both empirical results in [4] and in our work corroborate this assumption. To improve computational efficiency, **we avoid the calculation of second-order gradients** by stopping backpropagation through $\boldsymbol{\epsilon}_j$ in $\nabla \log p^*({\epsilon}_j)$ during the leapfrog steps $j \in \{1,.., L\}$, which sped up the execution time.
>
> Related to this comment, we added an additional Section A.5 showing the evolution of the step sizes $\boldsymbol{\phi}$ and the mean acceptance rate of the HMC sampler over the optimization steps. As observed in Figure 8, **thanks to our hyperparameter tuning, a mean acceptance rate of $\bar{p}_a=0.65$ is achieved at convergence**, which is considered as the optimal acceptance rate [1].
>
> > Limitations
>
> We believe that after addressing your comments, and comments from the rest of the Reviewers, the limitations of our proposed model are covered in the paper. To recap:
> * **Initialization of step sizes**.  We refer here to Figure 3 in [4], which is a similar HMC optimization approach. In the mentioned Figure, authors show the KSD between the HMC samples and the target distribution (the wave density illustrated in our experiment Figure 6 (a) and (b)) before and after training the hyperparameters for a large range of initialization step sizes. Authors demonstrate that \textbf{the method is robust to the initialization point provided it is not excessively large}, in which case every step in the chain is rejected and there is no gradient signal for learning. We experienced a similar behavior when the initial steps are too big, concretely when sampled from $U(0.2, 0.5)$.
>
> * **Initialization of the scaling factor**. Similarly, we show that the algorithm is robust when initializing $\boldsymbol{s}$ with a reasonable value. We choose an initial value of $s_l=1$ for all the scales, to start without scaling the variance. We observed empirically that in all cases the inflation increases over the training steps (as showed in new Figure 6 (c) and (f) of the revised paper).
> * **Choice of chain length $T$ and Leapfrog steps $LF$**. Bigger values lead to a more flexible posterior exploration with the disadvantage of an increased computational cost (as explained in Section 3.7).
> * **Depth of the hierarchy**. Too deep hierarchies are known to be problematic wrt to the optimization process, being posterior collapse of the deepest variables one of the typical issues to overcome. In our work, due to our motivation and the size of the benchmarks considered, we use a 2-layer hierarchy that was proved to be superior than a single-layered model.
>
> > Minor issues
>
> * We have extended Section A.1 with more experimental details to reproduce the results. In this work, we consider a 2-layer hierarchy. We plan to investigate on deeper hierarchies in future work and bigger datasets.
> * We corrected this notation by using $LF$ when referring to the number of Leapfrog steps. We would like to clarify that the length of HMC is given by $T$.
> * We agree that Eq. (7) is more appropriate. We have changed Eq. (5) by the ELBO of Eq. (7).
> * The typo is now solved in the revised version.
> * Solved in the revised version.
>
>
> ## References
>
> [1] M. Betancourt. A conceptual introduction to Hamiltonian Monte Carlo. arXiv preprint arXiv:1701.02434, 2017.
>
> [2] M. Betancourt and M. Girolami. Hamiltonian Monte Carlo for hierarchical models. Current
> trends in Bayesian methodology with applications, 79(30):2–4, 2015.
>
> [3] M. Girolami and B. Calderhead. Riemann Manifold Langevin and Hamiltonian Monte Carlo
> methods. Journal of the Royal Statistical Society: Series B (Statistical Methodology), 73(2):123–369
> 214, 2011
>
> [4] A. Campbell, W. Chen, V. Stimper, J. M. Hernandez-Lobato, and Y. Zhang. A gradient
> Based Strategy for Hamiltonian Monte Carlo Hyperparameter optimization. In International
> Conference on Machine Learning, pages 1238–1248. PMLR, 2021.

---

> ### Author Response · Authors · 2022-08-08
> **Rebuttal discussion**
>
> We would like to thank you for the time spent reviewing our paper. We would appreciate if you could let us know if our rebuttal and revised paper addressed your concerns and, if so if you could reconsider your rating for the paper. If you still have concerns, please let us know which ones these are, and we will try to address them again.

---

> > ### Comment · Reviewer_xZXW · 2022-08-09
> > **Post-rebuttal comment**
> >
> > Thank you for your comments on my questions and the review itself.
> > Here is my post-rebuttal comments.
> >
> > **On Q1. Difficulty or non-triviality in the combination of hierarchical model and HMC**
> >
> > ----
> >
> > > The naive application of HMC method over the hierarchical variables without reparameterization (Fig. 4a) does not work very well. When the states reached the mentioned problematic regions, the integrator diverged and we experienced overflow problems. By rejecting these states, chains got stuck and the hierarchical density was not properly explored. Consequently, HMC did not improve the Gaussian proposal. Given this pathological behavior, we consider this alternative an ill-posed model and it is not considered for evaluation/validation.
> >
> > Is it still pathological when the depth of the hierarchy is only two? I think adding the HH-VAEM without reparametrizaiton as a baseline in the experiment improves the objectivity of your claim.
> >
> > **On Q2. Computation of higher-order derivatives of HMC objective wrt the step sizes**
> >
> > ----
> >
> > > To improve computational efficiency, we avoid the calculation of second-order gradients by stopping backpropagation through  in  during the leapfrog steps , which sped up the execution time.
> >
> > If I understood it correctly, this is a heuristic that needs to be discussed in the theoretical viewpoint, but neither of this paper nor [4] did the right job in this regard. I would recommend the authors to be at least explicit on such heuristics in describing their algorithm.
> >
> > [4] A. Campbell, W. Chen, V. Stimper, J. M. Hernandez-Lobato, and Y. Zhang. A gradient Based Strategy for Hamiltonian Monte Carlo Hyperparameter optimization. In International Conference on Machine Learning, pages 1238–1248. PMLR, 2021.

---

> > > ### Author Response · Authors · 2022-08-09
> > > **Response to post-rebuttal comments**
> > >
> > > Thank you for your answer. Regarding the two post-rebuttal comments:
> > >
> > > > "Is it still pathological when the depth of the hierarchy is only two? I think adding the HH-VAEM without reparametrizaiton as a baseline in the experiment improves the objectivity of your claim."
> > >
> > > **When we tried to train HH-VAEM without reparameterization, HMC diverged and the model could not even been trained properly. Thus, including it as a baseline would not be possible**.
> > >
> > > It is well-studied in the literature that hierarchical models do not work properly with basic HMC implementations. When adding any number of AR variables (including a two-level depth), the hierarchical correlations make the HMC sampler pathological. The deeper the model is, the highest is the probability of falling into the aforementioned problematic regions where the integrator diverges. We reference the Reviewer to [2], where:
> > >
> > > * They show in Figure 2 that even a 2D hierarchical density provoke these issues.
> > > * In last paragraph of Section II.A, authors confirm that hierarchical models make naïve MCMC implementations become impractical, and that increasing the depth worsens this problem.
> > > * First two paragraphs in Section II.B, where authors propose the usage of auxiliary reparameterized variables for enabling Gibbs sampling to better explore the target.
> > >
> > >
> > > > About avoiding the calculation of second-order gradients: "I would recommend the authors to be at least explicit on such heuristics in describing their algorithm".
> > >
> > > **We will clearly state that heuristic when explaining the algorithm** in the camera-ready version of the paper.
> > >
> > >
> > > > Reconsideration of the rating
> > >
> > > We would like to ask you that, after we have addressed all the exposed concerns and improved our paper, and if you have no other concerns, you could reconsider updating your rating of the paper.
> > >
> > >
> > > ### References
> > >
> > > [2] M. Betancourt and M. Girolami. Hamiltonian Monte Carlo for hierarchical models. Current trends in Bayesian methodology with applications, 79(30):2–4, 2015.

---

> > > > ### Comment · Reviewer_xZXW · 2022-08-09
> > > > **Thank you for response**
> > > >
> > > > Thank you for your quick response.
> > > >
> > > > > When we tried to train HH-VAEM without reparameterization, HMC diverged and the model could not even been trained properly. Thus, including it as a baseline would not be possible.
> > > >
> > > > Thank you for the information. I think the fact that you have actually tried to train HH-VAEM without reparametrization and HMC diverged is important and I would recommend you to include it in the description of the baselines (Section 5) to explain why there is no such baseline.
> > > >
> > > > That being said, as you mentioned in Section 3.4, the training should be possible albeit the high probability of rejection. It is desirable to report the rejection rate or something to demonstrate its ill-posedness quantitatively.
> > > > Moreover, as for Fig 2 in [2], It is not obvious for me if the density landscape there is comparable to the autoregressive model in hierarchical VAEs enough for supporting the difficulty of the depth-2 model + HMC  in your experiment.
> > > >
> > > > Taking these into consideration, I will not change my score to express my concern with the objectivity of the key significance of the proposed method. **However**, my concern is now smaller than it was initially thanks to the authors and I do not mind if the paper is accepted when other reviewers argue for it.

---

> > > > > ### Author Response · Authors · 2022-08-09
> > > > > **Added new quantitative demonstration of the significance of the reparameterization contribution**
> > > > >
> > > > > In order to address your concern, we have included in the revised paper a new empirical demonstration of the significance of our reparameterization trick. **We have been able to add a new Section A.9 and Figure 10 that demonstrate quantitatively the significance of our novel contribution**.
> > > > >
> > > > > Following your suggestion, we included the variant of the model without reparameterization, i.e. directly inferring the autoregressive variables in Figure 4 (a). As we mentioned in Section 3.4, we reject the states that lead to overflow issues inside the integrator. **As expected, this lead to extremely low acceptance rates (we demonstrate this in new Figure 10 (a))**. On the contrary, using the reparameterization leads to a well-posed algorithm and valid acceptance rates. Further, we include the imputation metric in Figure 10 (b) showing that the ill-posed version gives much worse results, while our method successfully increases the metric.
> > > > >
> > > > > We hope that now your concern is fully solved, since we have strongly justified the significance of our contribution, as requested.

---

> > > > > > ### Comment · Reviewer_xZXW · 2022-08-10
> > > > > > **Thank you for the update**
> > > > > >
> > > > > > Yes, it resolves my concern. Now I would like to recommend the paper for acceptance.
> > > > > >
> > > > > > Thank you so much for addressing all of my concerns.

---

> ### Author Response · Authors · 2022-08-09
> **We hope all your post-rebuttal concerns are solved**
>
> We would like to thank you again for your time spent in providing a high-quality review of our paper, as well as your engagement during the rebuttal discussion. We hope that after **we have added your suggested demonstration for quantitatively justifying the significance of the reparamaterization method**, now all your concerns so far have been solved. For this reason, and if you do not have new concerns, we would like to encourage you to reconsider increasing the score accordingly.

---

### Official Review · Reviewer_Mwck · 2022-07-09

**Rating:** 6
**Confidence:** 3
**Soundness:** 3 good
**Presentation:** 4 excellent
**Contribution:** 3 good

**Summary:**

Uses Hierarchical VAE to impute missing heterogeneous data using HMC. Allows users to leverage a model with hierarchy and non Gaussians, which should lead to better modelling of the likelihood and generation.

**Questions:**

- The authors should add experiments focusing on high fidelity generation. Otherwise there doesn't seem to be much point in focussing on hierarchical VAEs if the results are presented on simple datasets. I'm aware that obtaining sota is often infeasible unless you work for FANG. But there should still be at least some effort to obtain results that match the motivation of the paper.
- If the authors can add additional datasets (I certainly don't expect these in two weeks) then I'm of the opinion that this paper should be accepted.

**Limitations:**

Doesn't seem to be.

**Strengths And Weaknesses:**

Strengths:
- The idea is novel and the goal is relevant to the literature.
- The approach is sound and suitable for the problem.
- The paper is well written and contains the relevant literature.

Weaknesses:
- The main weakness is that the results are not conclusive as the datasets used are limited.
- In the evaluation there doesn't seem to be a conclusion about why imputing data in hierarchal models is worth while? Hierarchical VAEs are renowned for high generation fidelity, so the authors should report results to highlight this.

---

> ### Author Response · Authors · 2022-08-02
> **Response to Reviewer Mwck**
>
> > The reviewer is ignoring all the motivations and contributions of our work.
>
> We would like to clarify the following: our paper presents two main contributions: i) HH-VAEM, a novel hierarchical VAE enhanced with automatically optimized HMC, which is evaluated on the tasks of missing data imputation, target prediction and active learning,  and ii) a novel Bayesian sampling-based framework for information acquisition that benefits from the design of HH-VAEM model. We provide the motivation and justification of our proposed design, along with empirical evidence that demonstrates the superiority of our proposed methods in the tasks considered.
>
> None of the motivations or contributions of our paper have been revised or even considered. The requested experiments are completely out of the scope and motivation of this work. Only by reading the title and abstract it remains clear that generation of high fidelity images is not a task we are considering in this paper. For this reason, the design choices of our model were not made to be evaluated in the requested task. We clearly motivate the usage of a hierarchical model based on the increased flexibility and inductive bias that it provides to the prior, and thus improving the performance in sampling-based tasks like the aforementeioned. We prove our hypothesis with experiments that demonstrate the superiority of the hierarchical models in the three considered tasks, which we clearly enumerate in the abstract and repeatedly in the paper: i) missing data imputation, ii) target prediction and iii) active information acquisition.
>
> Apart from this consideration, we believe that the datasets employed are not limited, since they are widely employed as benchmarks for the considered tasks.

---

> > ### Comment · Reviewer_Mwck · 2022-08-03
> > **1st Comment**
> >
> > I would like to highlight that I am not 'ignoring all the motivations and contributions' of your work. And please forgive me, but my understanding of the goal of the work is to perform data imputation? Surely then it is reasonable to ask to see what imputations the model produces? Or have I completely missed the point?
> >
> > Indeed, upon reflection of my review and re-reading the paper and the related literature it seems that my requests of additional high fidelity experiments are unjust; and I have updated my scores in response. However, assuming my above assumption is correct, I still feel that the paper could benefit from further experiments; some qualitive examples for MNIST would be nice to see for a start (I see them in [46]). I would also imagine that adding results for CelebA wouldn't be too complicated either, although I'm aware that adding these over the review period is infeasible and in the event that this paper is not accepted, I would encourage the authors to add these.

---

> > > ### Author Response · Authors · 2022-08-07
> > > **Added inpainting experiments with CelebA and MNIST**
> > >
> > > We would like to thank you for re-reading the paper and reconsidering the score. We are glad to announce that, following this comment,  we have improved our paper by introducing the requested experiments on image imputation in the revised version (new Section A.8 and Figure 9). Concretely, we have trained our model and baselines on CelebA, and perform the **evaluation on conditional image inpainting on both CelebA and MNIST**.
> > >
> > > Thanks to the design of our model, we can encode the observed part of the image to obtain the approximate posterior $q(\boldsymbol{z} | \boldsymbol{x}_O)$ (or $q(\boldsymbol{\epsilon} | \boldsymbol{x}_O)$ for the hierarhical models). Samples from this posterior are decoded to impute the missing part using $p(\boldsymbol{x}_U | \boldsymbol{z})$ (or $p(\boldsymbol{x}_U | \boldsymbol{h}_1)$). We expect a qualitative improvement in terms of imputation after introducing the hierarchy for increasing the flexibility of the model. Further, similarly than for all the previously considered tasks, this task requires sampling from the approximate posterior of the model. For this reason, we again motivate the usage of our HMC-based model for obtaining better imputations due to providing better samples by HMC (Figure 1b). New results in new Section A.8 visually demonstrate these gains, and include analysis and discussions of the imputation provided by our model and the baselines (Figure 9).
> > >
> > >
> > > We hope that after following your suggestions and adding the aforementioned results, your concerns have already been solved for accepting the paper.

---

> > > ### Author Response · Authors · 2022-08-08
> > > **Reconsideration after adding requested experiments**
> > >
> > > Thanks for the time spent reviewing our paper. We would appreciate if you could let us know if our newly added experiments with CelebA and MNIST addressed your concerns and, if so if you could reconsider increasing your rating for the paper. If you still have concerns, please let us know which ones these are, and we will try to address them again.

---

> ### Author Response · Authors · 2022-08-09
> **We hope all your concerns are solved**
>
> We would like to thank you again for reviewing our paper. We hope that after **we have added the requested experiments on image imputation with CelebA, demonstrating the significance of our design choices and the superiority of our model**, now all your exposed concerns have been solved. For this reason, we would like to encourage you to reconsider increasing the score accordingly.

---

### Official Review · Reviewer_Uewz · 2022-07-13

**Rating:** 6
**Confidence:** 4
**Soundness:** 2 fair
**Presentation:** 3 good
**Contribution:** 3 good

**Summary:**

The authors combine hierarchical VAEs and MCMC-within-VI, while emphasizing that the hierarchical aspect of VAEs is useful for dealing with missing data problems.

**Questions:**

1.  In lines 73-74, "unobserved data is replaced with zeros" and line 139, "using zero-filling for the unobserved variables"

Whether or not zero imputation leads to lower bounds on observed data likelihoods  depends on assumptions on the missing data mechanism. For example, [31] make the missing at random (MAR) assumption. However, there is not a discussion of assumptions on misingness in your text. Since it is impossible to work with missing data without assumptions [see e.g. Tsiatis Semiparametrics and Missing Data], which assumptions do you make on missingness in your text?

In the experiments you describe :

"For all the models, we manually introduce missing data in the training set by randomly setting per data point a feature as missing with a probability sampled uniformly in the interval [0.01, 0.99] within each batch. Both the input data x and the target y can be missing.

This means the x dimensions are "missing completely at random". Which modifications to your method are necessary if some dimensions of x are missing conditional on y or on other dimensions of x? If you don't handle this case in your work, you should state that.


2. For the predictive model ptheta(y|xhat, h), could you give a precise equation or sampling statement for xhat?

I get roughly that you encode x (with some features possibly missing) into the per-dimension z's, and from those to h, then decode from h to z's, and finally "each marginal z_d into xhat_u".

How exactly does the last step "each marginal z_d into xhat_u" work? Does each z_d generate dimension d of xhat_u? For those x features that are observed, do you set xhat_d to x_d?

3. (lines 219-223) "Nonetheless, by using our reparameterized version, we are able to relax these dependencies and maintain the standard HMC method for obtaining samples that approximately follow the true posterior."

How was this evaluated/measured? Test NLLs of the resulting models (for x and for y) are a good goal in general for modeling but these NLLs do not necessarily show that inference worked in the way that is claimed.


4.  You report test NLLs for missing features and for y, but how about held-out test NLL of the observed features? Whether or not it is also necessary to report NLLs on observed features again depends on the missingness assumptions. Also, possibly discrepancies between likelihoods on observed and missing data dimensions may depend on how hard the modeling tasks for those two sets of features are.
Since you only implicitly described the missingness assumptions by describing the algorithm, it would be helpful for you to state what about your missingness assumptions and general data generation assumptions justifies the choice of reported metrics. Without further assumptions stated, I would wonder about held-out likelihood on observed data dimensions.

5. What do you do with target prediction NLLs for data that has missing y? Exclude?


6. The technical part of this paper seems to be about training (a variant of) hierarchical VAEs with MCMC and with a novel 2-part ELBO, rather than a specific paper on missingness (since missingness assumptions are not really discussed). On the other hand, the experiments highlight missingness. I would possibly suggest changing the title as well as revise the text to mention more missingness.


7. A writing suggestion: never end sub-sections with an equation (e.g. sections 3.2 and 3.4). It helps to make a concluding remark.

I am tentatively scoring "weak accept" due to good results but confusing discussion of assumptions and metrics. Glad to improve my score if authors sufficiently engage with these questions and revise the text.



**Limitations:**

Yes.

**Strengths And Weaknesses:**

Strength:

Novel combination of hierarchical VAEs (modeling flexibility) and MCMC-within-VI (inference quality) along with a new objective that addresses two distinct ELBOs of the model.

Weaknesses:

some assumptions/scope not clear. See Questions.

---

> ### Author Response · Authors · 2022-08-02
> **Response to Reviewer Uewz (part 1/2)**
>
> We thank the reviewer for the positive feedback and detailed suggestions. Below, we address each of your comments:
>
> > Which assumptions do you make on missingness in your text? Which modifications to your method are necessary if some dimensions of x are missing conditional on y or on other dimensions of x?
>
> Following your suggestion, **we have added new references in 2.1 and clarified in Section 3.2 that we are using the missing-at-random assumption (MAR)**, similarly than in [1]. We are not implicitly modeling missingness, since our model generates complete data. Nevertheless, the encoder input is fed with concatenated masks containing the missing positions for both input and target, thus, the posterior distribution capture missing patterns for decoding data. In future work more focused on image generation with bigger datasets, we plan to investigate how the decoded images from the posterior change when using different missing patterns at the input.
>
> > Precise equation or sampling statement for the input of the predictor $\hat{x}$
>
> **We have added extra notation for clarifying the input of the predictor** in Section 3.3. Namely, $\hat{\boldsymbol{x}}$ is the vector containing the observed variables $\boldsymbol{x}_O$ and the imputation of the missing variables $\hat{\boldsymbol{x}}_U$:
> $$\hat{\boldsymbol{x}} = \left( x_i \in \boldsymbol{x}_O, \; \hat{x}_j \in \boldsymbol{x}_U \right) $$
>
> where $\hat{x}_j$ is obtained from the corresponding marginal VAE $p(x_j|z_j)$, and $z_j$ corresponds to the $j$-th dimension of $\boldsymbol{z}$, decoded from the dependency VAE $p(\boldsymbol{z | \boldsymbol{h}_1})$.
>
> > How was the reparameterization version evaluated/measured?}
>
> **We have reorganized Section 3.4 in order to highlight that the alternative of not using the reparameterization is ill-posed and thus it is not considered for evaluation/validation**. As stated in [2, 3], HMC can be pathological when used for sampling from hierarchical densities, where huge autoregressive variations increase exponentially with the depth (check simple 2D funnel example on Figure 3 in [2]). For approximating the Hamiltonian dynamics, inside each Leapfrog integrator step, gradients $\nabla_{h_{1:L}} \log p^*(h_{1:L})$ are required. Due to the strong curvature regions, high norm gradients are backpropagated and might eventually explode, ending in overflow issues (Fig. 35 in [1]).
>
> We firstly tried running our HMC method over the hierarchical variables without reparameterization (Fig. 4a). When the states reached the mentioned problematic regions, the integrator diverged and we experienced overflow problems. By rejecting these states, chains got stuck and the hierarchical density was not properly explored (thus, HMC did not improve the Gaussian proposal).
>
> Instead of dealing with advanced versions of HMC, e.g. Riemann manifold HMC [4] with momentum variance depending on the state position, we successfully solved this by using our proposed reparameterization design.  Thanks to this design choice, we demonstrate with results that HMC is effective and superior in the considered tasks wrt to the Gaussian alternatives.

---

> ### Author Response · Authors · 2022-08-02
> **Response to Reviewer Uewz (part 2/2)**
>
> > Held-out test NLL of the observed features
>
> After clarifying that we are using the missing-at-random assumption in the paper, and following your suggestion, **we have included the following Table of test NLL on observed features (see Section A.6, Table 7)** in the revised version of the paper. Thanks to this experiment, we observe that: i) all HMC-based models outperform Gaussian models in reconstruction, and ii) although the enriched hierarchical prior improves the missing data imputation, it does not necessarily beat the one-layered alternative in all the cases in terms of reconstruction.
>
> |         |                 bank |            insurance |              avocado |                naval |               yatch |             diabetes |             concrete |                 wine |               energy |               boston |
> |---------|---------------------:|---------------------:|---------------------:|---------------------:|--------------------:|---------------------:|---------------------:|---------------------:|---------------------:|---------------------:|
> | VAEM    |      $0.51 \pm 0.05$ |      $0.99 \pm 0.05$ |      $0.44 \pm 0.01$ |      $0.21 \pm 0.01$ |     $0.62 \pm 0.13$ |      $0.92 \pm 0.12$ |      $0.63 \pm 0.18$ |      $0.73 \pm 0.18$ |      $1.86 \pm 0.09$ |      $0.56 \pm 0.11$ |
> | MIWAEM  |      $0.63 \pm 0.02$ |      $1.06 \pm 0.03$ |      $0.60 \pm 0.03$ |      $0.33 \pm 0.01$ |     $0.75 \pm 0.07$ |      $1.05 \pm 0.06$ |      $0.76 \pm 0.09$ |      $0.80 \pm 0.06$ |      $1.77 \pm 0.15$ |      $0.67 \pm 0.03$ |
> | H-VAEM  |      $0.40 \pm 0.04$ |      $0.93 \pm 0.04$ |      $0.42 \pm 0.05$ |      $0.19 \pm 0.07$ |     $0.58 \pm 0.09$ |      $0.70 \pm 0.13$ |      $0.53 \pm 0.18$ |      $0.71 \pm 0.15$ |      $0.38 \pm 0.02$ |      $0.49 \pm 0.07$ |
> | HMC-VAE |      $0.37 \pm 0.07$ | $\boldsymbol{0.92 \pm 0.04}$ |      $0.39 \pm 0.06$ |      $0.18 \pm 0.05$ |     $0.54 \pm 0.10$ | $\boldsymbol{0.68 \pm 0.07}$ |      $0.49 \pm 0.22$ | $\boldsymbol{0.55 \pm 0.07}$ |      $0.40 \pm 0.06$ |      $0.41 \pm 0.04$ |
> | HH-VAEM | $\boldsymbol{0.33 \pm 0.03}$ |      $0.95 \pm 0.05$ | $\boldsymbol{0.36 \pm 0.01}$ | $\boldsymbol{0.17 \pm 0.04}$ | $\boldsymbol{0.45\pm 0.04}$ |      $0.68 \pm 0.16$ | $\boldsymbol{0.40 \pm 0.16}$ |      $0.64 \pm 0.17$ | $\boldsymbol{0.37 \pm 0.06}$ | $\boldsymbol{0.41 \pm 0.04}$ |
>
>
> > What do you do with target prediction NLLs for data that has missing y? Exclude?
>
> Missing data is manually dropped out using artificial missing masks over the originally complete datasets. For the test target, the mask is filled to be fully missing, since we evaluate our method in prediction tasks when using only the input data $\boldsymbol{x}_O$. When computing the NLL, we use the original test values as ground truth to evaluate how likely are the predicted values.
>
>
> > I would possibly suggest changing the title as well as revise the text to mention more missingness
>
> Considering that i) partial data is key in the design choices of the model, ii) imputing missing data is involved in all the tasks considered in this work, and iii) we show the superiority of our design in all this tasks, we decided to maintain the title. After carefully reading your comments, we have improved and extended the analysis of the missingness assumptions. Thanks to this, we believe the revised paper fits better to the title.
>
> > Never end sub-sections with an equation
>
> We reorganized the end of sections 3.2 and 3.4 according to your suggestion.
>
>
> ## References
>
> [1] P.-A. Mattei and J. Frellsen. MIWAE: Deep generative modelling and imputation of incomplete
> data sets. In International Conference on Machine Learning, pages 4413–4423. PMLR, 2019.
>
> [2] M. Betancourt. A conceptual introduction to Hamiltonian Monte Carlo. arXiv preprint arXiv:1701.02434, 2017.
>
> [3] M. Betancourt and M. Girolami. Hamiltonian Monte Carlo for hierarchical models. Current
> trends in Bayesian methodology with applications, 79(30):2–4, 2015.
>
> [4] M. Girolami and B. Calderhead. Riemann Manifold Langevin and Hamiltonian Monte Carlo
> methods. Journal of the Royal Statistical Society: Series B (tatistical Methodology), 73(2):123–369
> 214, 2011

---

> ### Author Response · Authors · 2022-08-08
> **Rebuttal discussion**
>
> Thanks for the time spent reviewing our paper. We would appreciate if you could let us know if our rebuttal and revised paper addressed your concerns and, if so if you could reconsider your rating for the paper. If you still have concerns, please let us know which ones these are, and we will try to address them again.

---

> ### Author Response · Authors · 2022-08-09
> **Rebuttal discussion reminder**
>
> We would like to remind you that after we have addressed all your concerns so far, we are still open to address more if you still have any. In case not, we ask you to consider increasing the score from 6 Weak accept to a higher score, as mentioned in your review.

---

### Official Review · Reviewer_jW2a · 2022-07-13

**Rating:** 6
**Confidence:** 4
**Soundness:** 3 good
**Presentation:** 2 fair
**Contribution:** 3 good

**Summary:**

The authors propose a new way of doing generative modeling  with an incomplete (potentially labelled) heterogeneous data set. Dealing generatively with heterogeneous and/or incomplete data is an important open problem, for which few solutions exist today.

Their approach is to construct a generative model based on a new kind of hierarchical VAE (HVAE) architecture. The model is trained via a blend of variational inference and Hamiltonian Monte Carlo (HMC).

After training, they mainly look at two specific applications: imputation and active learning of the missing values. Regarding active learning, they propose a new approach to estimate the mutual information using HMC samples, improving on the previous approach of Ma et al. (2019) to using VAEs for active learning of the missing values.

They perform a series of experiments that indicate that their HVAE outperforms other VAE-type models for heterogeneous data, for imputation, prediction of the labels, active learning, and anomaly detection.


Ma et al., EDDI: Efficient Dynamic Discovery of High-Value Information with Partial VAE, ICML 2019

**Questions:**

My questions are contained in the "Weaknesses" section of my review.

**Limitations:**

I do not see potential negative societal impact that this work could create.

About the limitations, I think the authors should discuss the assumptions necessary for their method to work. In particular, the fact that they consider the likelihood of the observed data suggests that their method requires a missing at random assumption. This should be discussed. On the other hand, I really like that the authors include computing times.

**Strengths And Weaknesses:**

STRENGTHS

The paper has many clever contributions. I list and praise them below.

1) The idea called "predictive enhancement" by the authors is a quite smart blend of generative and discriminative ideas. Indeed $p(x,y)$ is modeled as $\int p(z)p(x|z)p(y|z,x)dz$, where $p(x|z)$ is parametrised by a generative network, and $p(y|z,x)$ by a discriminative one.

2) The HVAE architure proposed by the authors is quite different from standard ones, as it models directly the "noise" variables implied in the reparametrisation trick. This in particular allows the authors to avoid complex inference network architectures used in other HVAE papers.

3) The authors propose a new way to perform HMC in a VAE context, by focusing on the "noise" variables $\epsilon$, which do not depend on any parameters, and then doing variational inference where the proposal for $\epsilon$ is obtained through HMC. Because of HMC, the likelihood lower bound can't be estimated unbiasedly, so they use a few tricks to transform it into something that can be optimised.

4) I like the extended scope of the experiments, even if I have a few issues with some experimental details.


WEAKNESSES

The main weakness of the paper is that these individual contributions are described too hastily, and without much discussing of the motivations, what is new, what is not, and what is inspired by other work. More importantly, even though several interesting ablation studies are done, it remains unclear what contributions bring the most to the table.

a) It is not clear if there is anything new in Section 3.2, compared to Ma et al. (2020, cited as [28]). If there isn't, why include this here rather than in the "related work" section?

b) As I mentioned above, the "predictive enhancement" idea is quite clever, and likely to improve the quality of the predictions substantially. I find it however quite odd not discussing more its motivations or related works. In this missing-data context, using a generative model combined with a discriminative one goes back at least to the work of Tresp et al. (1994) and Ghahramani and Jordan (1995), who used a GMM for $p(x)$ instead of a VAE. In a deep learning context, this idea was also explored by Smieja (2018) and by and Ipsen et al. (2022). Supervised VAE models have also used similar factorisations, for instance Joy et al. (2021) essentially uses a factorisation of type $p(z)p(x|z)p(y|z)$, and the DFZ model of Li et al. (2019) uses the same factorisation as you, i.e. $p(z)p(x|z)p(y|x,z)$.

c) The small change in HVAE architecture is poorly motivated (Section 3.4). Many claims here are vague and not really supported by theory or evidence, e.g. "However, by choosing this reparameterization, we relax the dependencies among the latent variables, resulting in a smoother joint posterior density p(ϵ|xO, yO)" (what does smoother means here ? Maybe doing a toy experiment would help ?). Is it only useful for HMC or does it also change something for variational inference? How does it relate to usual tricks for HVAEs?

d) I must admit I do not understand the rationale behind the modification of the ELBO proposed in Section 3.5. The problematic term $H(q_\phi)$ is simply removed while the variance of $q_\phi$ is artificially inflated (the authors do not specify exactly how) by a hyperparameter $\mathbf{s}$. This $\mathbf{s}$ is fit by minimising a divergence (the Sliced Kernelized Stein Discrepancy) between the true posterior and $q_\phi$. This trick needs to be more properly explained, and maybe illustrated on a simple example. How does this trick relates to other methods that have been used to do HMC in a VAE context, for instance Caterini et al. (2018 [8]) or Hoffman (2017 [19]) ?

e) While nicely motivated, I think that the description of the mutual information estimate of Equation (13) is a bit unclear: which samples are used ? how do you create the bins? What are the properties of the estimate?

I also have a few issues with the imputation experiments:

f) To evaluate the quality of the imputations, you approximate the test log likelihood of the missing values. Equation (14) reads $ \log p(x_U | x_O) = \log E_{\epsilon \sim q} p(x_U | \epsilon) \approx log \frac{1}{k} \sum_{i=1}^k p(x_U | \epsilon_i)$,
but the first part of the equation is false because $q$ is not equal to the true posterior. I think this evaluation protocol is sensible, but requires a lot of approximations that can be hard to grasp (for instance, it is possible that the models trained with HMC are not really "better", but leads to posteriors that are better approximated by HMC, so a HMC-based evaluation of non-HMC-based models may be seen as a bit unfair). I think it would be nice to keep this evaluation method, but maybe complement it with something simpler, for instance the imputation error (MSE or accuracy). Another nice thing with the imputation error is that it allows you to easily add non-VAE baselines, which would add a lot of information. For instance, for heterogeneous data, the missforest R package (based on Stekhoven and Buehlmann, 2012) is still, in my opinion, easy to use and very difficult to beat. It would be interesting to see how it compares to VAE-like models.

g) Moreover, as you rightfully claim in the beginning of the paper, in an heterogeneous context, "the problem of handling unbalanced likelihoods leads to the domination of some dimensions during the optimization process". This means that the full likelihood can be a bit misleading, and showing feature-specific scores (maybe in the appendix) may be interesting.

MINOR ISSUES



- There are a few issues in the references. Most of them are in a numeric format, but some are not, e.g. (Betancourt et al., 2015) line 165. In multiple cases of published papers, the arxived versions are cited instead of the published ones (e.g. Hendrycks and Dietterich was published at ICLR 2019, the two papers of Ma et al. were published at ICML 2019 and NeurIPS 2020, Maaløe et al. was published at NeurIPS 2019, Caterini et al. was published at NeurIPS 2019).

- sksd is never properly defined



ADDITIONAL REFERENCES

Ghahramani and Jordan. Learning from incomplete data. MIT Center for Biological and Computational Learning Technical Report 108, 1995.
Joy et al., Capturing Label Characteristics in VAEs, ICLR 2021
Li et al. Are Generative Classifiers More Robust to Adversarial Attacks?, ICML 2019
Tresp et al. Training neural networks with decient data. NeurIPS 1994
Smieja et al., Processing of missing data by neural networks, NeurIPS 2018
Ipsen et al., How to Deal with Missing Data in Supervised Deep Learning, ICLR 2022
Stekhoven and Buehlmann, MissForest - nonparametric missing value imputation
for mixed-type data, Bioinformatics, 2012




---------------- POST-REBUTTAL EDIT ------------------

Many thanks for this thorough revision. A lot of these updates are significant, in particular the additions of new experiments (more proper metrics, adding missforest, doing impaining), and the clarifications of the active learning strategies.

My last remaining significant concern is the lack of strong justification for the HMC tricks that the authors use (the variance inflation, the SKSD). I appreciate the additional details given by the authors though, and I understand the general rationale.

Overall, I am updating my score to 6, since I think this paper would be a valuable fit for NeurIPS.

---

> ### Author Response · Authors · 2022-08-02
> **Response to Reviewer jW2a (part 1/3)**
>
> Thank you for the constructive feedback and detailed suggestions. Below, we respond to each of your comments:
>
> > Motivations, contributions and inspirations
>
> Following your consideration, **we have strengthened the description of the motivations, contributions and inspiration on other works**.
> We clarify in the revised version that:
>
> * The goal of this work is to improve the alternatives in the three considered tasks: missing data imputation, target prediction and active learning. All of them would benefit from models with a flexible prior and that provide samples approximately following the true posterior. More specifically, a sampling-based approach for information acquisition would specially benefit form better quality samples.
> * First, inspired by previous work on hierarchical VAEs, we decided to increase the flexibility of the prior by adding a hierarchy of latent variables. Second, inspired by previous work on optimizing the hyperparameters of HMC for sampling from the posterior of a VAE, we decided to use a similar approach for sampling from our hierarchical posterior. Third, inspired by simple but effective mutual information estimators, we created a novel sampling-based method for obtaining an information reward using histograms.
> * Learning HMC hyperparameters on a hierarchical autoregressive density is not a well-posed problem. We achieve a well-posed solution by using our proposed reparameterization method.
> * We provide evidence in the experiments that both the reparameterized hierarchy and the tuned HMC incrementally improve the alternatives (one-layered VAEs, Gaussian-based encoders) in the tasks considered. Our sampling-based approach is also proved to be superior to the Gaussian-based alternative.
>
> > Novelty compared to Ma et al. (2020, cited as [28])
>
> Our novelty lies in the re-design of the dependency VAE component of Ma et al. (2020, cited as [28]), so that its synergy to HMC inference is improved. **We are the first to do HMC with hierarchical VAEs**, providing a new parametrization of the hierarchical VAE that makes HMC work with such models by reducing strong correlations in the posterior distribution of latent variables (this is extensively detailed in section "Motivation of the reparameterization technique" below).
>
> > References and motivation on the "Predictive enhancement" approach
>
> Following this comment, **we have added a new paragraph in Section 3.3 including all the mentioned references and motivating the combination of generative and supervised models**.
>
> > Motivation of the reparameterization technique
>
> Considering this comment and similar comments from other reviewers, **we have reorganized Section 3.4 for strengthening the motivation of our proposed reparameterized hierarchy**. As stated in [1, 2], HMC can be pathological when used for sampling from hierarchical densities, where autoregressive variations increase with the depth (check simple 2D funnel example on Figure 3 in [2]). For approximating the Hamiltonian dynamics, inside each Leapfrog integrator step, gradients $\nabla_{\boldsymbol{h}_{1:L}} \log p^* ( \boldsymbol{h} )$ are required. Due to the strong curvature regions, high norm gradients are backpropagated and might eventually explode, ending in overflow issues (Fig. 35 in [1]).
>
> We firstly tried running our HMC method over the hierarchical variables without any reparameterization (Fig. 4a). By the time the states reached the mentioned problematic regions, the integrator diverged and we experienced the aforementioned overflow problems. By rejecting these problematic states, chains got stuck close to the proposal and the hierarchical density was not properly explored (thus, HMC did not improve the Gaussian proposal). This version is not included in the study, since we considered this alternative a pathological procedure.
>
> We successfully solved this issue by using our proposed reparameterization technique. The prior $p(\boldsymbol{\epsilon_1}, ..., \boldsymbol{\epsilon_L}) =  \prod_{l=1}^L p(\boldsymbol{\epsilon_l})$ is no longer autoregressive, thus, the posterior is smoother (do not contains huge variations) and the derivatives computed in the Leapfrog steps of HMC are relaxed.

---

> ### Author Response · Authors · 2022-08-02
> **Response to Reviewer jW2a (part 2/3)**
>
> > A more detailed explanation on the HMC optimization method.
>
> **We have extended the details of the HMC optimization method in Section 3.5**. **Further, we have added a toy experiment A.3 showing the efficacy of the HMC optimization algorithm**.
>
> Inpired by [3], we use a VI approach for optimizing the hyperparameters of HMC, being $q_\phi^{(T)}$ our variational approximation of $p^*$. The entropy term  of ELBO in Eq. (8) acts as a regularizer preventing $q_\phi^{(T)}$ from simply collapsing to a point mass at the mode of $p^*$.
> Nevertheless, this term is not possible to compute, since $q_\phi^{(T)}$ is the implicit distribution of the HMC samples at state $T$.  Thus, not considering the entropy term does have some implications. In case the initial distribution, $q_\phi^{(0)}$, were concentrated in a very high probability region of the target, $p^*$, uniquely optimizing the first term would encourage the chains to stay close to their initial sampling point, thus not exploring the full target, which is undesirable
> behaviour. The key for preventing this issue is choosing an initial distribution that sufficiently covers the target, which we achieve by inflating the proposal by an optimized parameter.
>
> The way we achieve this regularization is by optimizing the scaling parameter $\textbf{s}$ for inflating the proposal $q_\phi^{(0)}$. We inflate the variance at each layer (as specified in the revised paper).
> and $\textbf{s}=\{s_1, ..., s_L \}$ is tuned by ensuring that the inflated modification better covers the target, which is obtained by minimizing the SKSD.
>
> This trick is not related to any of the mentioned HMC methods in VAE context, since they do not automatically optimize the HMC hyperparameters using a VI approach, like we propose in our work. We were inspired by [3], were a more detailed discussion can be found. Further, related to this comment, we have added another experiment A.4, demonstrating the efficacy of the algorithm by showing that the mean acceptance rate converges to the optimal value of $\bar{p}_a = 0.65$ [4].
>
> > Clarifying the description of the MI estimate
>
> **We have included a more detailed description in Section 4**. Rather than using KLs on the approximated Gaussian distributions given by the encoder, as in [4], we use samples from the posterior, provided by the optimized HMC, thus expecting to improve the data acquisition by overcoming this bias. These samples are decoded to obtain both variables $x_d$ and target $y$. The number of bins is a hyperparameter, which defines the resolution of the histograms, i.e. uniformly distributed intervals over $x_d$ and $y$ supports. In case the feature or the target is discrete, we use the categories as bins. Since this estimator is sampling-based, under certain conditions (namely,  if all densities exist as proper functions), Eq. (13) indeed converges to $\mathcal{I} (\boldsymbol{y} ; x_i  \; | \boldsymbol{x}_O)$ if we first let the number of samples $N \to \infty$ [5].
>
> > On the evaluation of missing data imputation. Include the imputation error and compare with non-VAE baselines like missForest.
>
> We have corrected the typo by changing $=$ by $\approx$. We would like to clarify that Equation (14) is not a HMC-based evaluation, since it depends on the posterior approximation provided by the model. In the case the posterior approximation is restricted to be Gaussian ($q^{(0)}$), HMC is not used. Our proposed model outcomes this restriction, obtaining approximations that closely follow the true posterior (Figure 1b).
> Following your suggestion, **we have included Table 6 in section A.3 of the paper, with the suggested metrics for continuous/classification tasks, including the missForest package as a new baseline**. We propose to change the accuracy by the error rate in order to provide an average metric for all the imputed variables. The table shows that, in almost all the cases, HH-VAEM beat all the alternatives and baselines, and the incremental improvement when adding the hierarchy and HMC is achieved.

---

> ### Author Response · Authors · 2022-08-02
> **Response to Reviewer jW2a (part 3/3)**
>
> > The full likelihood can be a bit misleading
>
> We agree with that the reported average likelihood across heterogeneous dimensions is not a perfect metric. The reason why we show the average is to provide a fair comparison on models that have been trained on the same heterogeneous likelihoods. We have analyzed this at the beginning of Section 5. Further, following your suggestion, **we included in the Appendix A.7 tables 8-10 reporting average likelihoods on the three data-types considered for some datasets**, showing again the incremental improvement when adding the different parts considered in our proposal.
>
> > Minor issues and limitations
>
> All the minor issues are now solved. Additionally, we have clarified in the text of the paper that we make use of the missing-at-random assumption.
>
> ## References
>
> [1] M. Betancourt. A conceptual introduction to Hamiltonian Monte Carlo. arXiv preprint arXiv:1701.02434, 2017.
>
> [2] M. Betancourt and M. Girolami. Hamiltonian Monte Carlo for hierarchical models. Current
> trends in Bayesian methodology with applications, 79(30):2–4, 2015.
>
> [3] A. Campbell, W. Chen, V. Stimper, J. M. Hernandez-Lobato, and Y. Zhang. A gradient
> Based Strategy for Hamiltonian Monte Carlo Hyperparameter optimization. In International
> Conference on Machine Learning, pages 1238–1248. PMLR, 2021.
>
> [4] R. M. Neal et al. MCMC using Hamiltonian dynamics. Handbook of Markov Chain Monte Carlo,
> 2(11):2, 2011.
>
> [5] A. Kraskov, H. Stögbauer, and P. Grassberger. Estimating mutual information. Physical review
> E, 69(6):066138, 2004.

---

> ### Author Response · Authors · 2022-08-08
> **Rebuttal discussion**
>
> Thanks for the time spent reviewing our paper. We would appreciate if you could let us know if our rebuttal and revised paper addressed your concerns and, if so if you could reconsider your rating for the paper. If you still have concerns, please let us know which ones these are, and we will try to address them again.

---

### Author Response · Authors · 2022-08-08
**Summary of the changes in the revised version**

Dear reviewers,

We would like to thank you for your time considered to review our paper and provide insightful feedback. We have addressed all your concerns and answered personally to each of your reviews. We uploaded a new revised version of the paper that includes all the suggested improvements, experiments, validations and modifications. We encourage discussion on the rebuttal and revised paper, in order to address more concerns in case you still have any. We hope that after we have addressed all your concerns for improving our paper, you could reconsider your rating.

We list below the summary of major changes introduced:

* **New toy experiment A.3** **illustrating the efficacy of training the HMC hyperparameters on 2D-densities** (Figure 6), for extending the explanation of the HMC method, as suggested by *Reviewer jW2a*.
* **New experiment A.4** with the **acceptance rate and steps sizes of HMC over the optimization steps** (Figure 8), for a more detailed discussion on the practical considerations of the HMC sampler, as suggested by *Reviewer gfdz*. This experiment is also considered an extension on the HMC method explanation suggested by *Reviewer jW2a*.
* **New experiment A.5** computing the **imputation error metrics**, and comparing with the missForest baseline (Table 6), as suggested by *Reviewer jW2a*.
* **New experiment A.6** computing the **likelihood of the observed features** (Table 7), as suggested by *Reviewer Uewz*.
* **New experiment A.7** computing the **likelihood for each data-type** separately (Tables 8-10), as suggested by *Reviewer jW2a*.
* **New experiment A.8** qualitatively evaluating our model in **conditional image inpainting on CelebA and MNIST** (Figure 9), as requested by *Reviewer Mwck*, showing the superiority of our proposed model.
* To address comments from *Reviewers jW2a*, *Uewz* and *xZXW*, **Section 3.4 is rewritten** for clarifing the non-triviality of combining HMC with a deep hierarchical generative model, and highlighting that **we are the first to successfully solve the pathological issues of combining HMC and hierarchical VAEs by using our proposed reparameterization method**.
* **We have strengthened the discussions and validations** of introducing each design choice of our HH-VAEM and our information acquisition method, as suggested by *Reviewer xZXW*.
* **We have extended the discussion and analysis of all the results**, as suggested by *Reviewer xZXW*.
* **We have clarified in the text of the paper that we make the missing-at-random (MAR) assumption**, as suggested by *Reviewers jW2a* and *Uewz*.

---

> ### Comment · Reviewer_jW2a · 2022-08-08
> **Thanks!**
>
> Many thanks for this thorough revision. A lot of these updates are significant, in particular the additions of new experiments (more proper metrics, adding missforest, doing impaining), and the clarifications of the active learning strategies.
>
> My last remaining significant concern is the lack of strong justification for the HMC tricks that the authors use (the variance inflation, the SKSD). I appreciate the additional details given by the authors though, and I understand the general rationale.
>
> Overall, I am updating my score to 6, since I think this paper would be a valuable fit for NeurIPS.

---

> > ### Author Response · Authors · 2022-08-08
> > **Regarding last concern on HMC optimization tricks**
> >
> > Many thanks for your positive feedback and update of the score. Regarding your last concern, we believe that **new section A.3 added in the revised version already addresses this and justifies why inflating the variance with a tunable parameter is crucial for a better HMC optimization**. Concretely, Figure 6 perfectly illustrates this. In Figure 6 (c), the inflation parameters (per dimension) are showed over the training steps. Minimizing the SKSD makes that $s_1$ increases to inflate the horizontal variance of the Gaussian proposal. Thanks to this, the tight initial proposal is inflated to a better distribution that widely covers the objective for initializing the HMC chains. Further, Section A.4 and Figure 8 also justify the efficacy of the trick.
> >
> > Regarding the justification for using the SKSD discrepancy, we remark that it perfectly fits for our algorithm, due to the following reasons:
> > * Similarly than a KL divergence does in VI, it acts as a regularization term. As an example, one can think on a multimodal distribution. It penalises the non-desired behaviour of HMC samples getting stuck in one mode of the density. On the contrary, it is minimised when the proposal widely covers the density and HMC samples reach all the modes.
> >
> > *  Its computation requires gradients $\nabla_{\boldsymbol{z}} \log p(\boldsymbol{z}| \boldsymbol{x})$ (or $\nabla_{\epsilon} \log p( \boldsymbol{\epsilon} | \boldsymbol{z}, \boldsymbol{y})$ for our model) of the objective and samples of the approximation $q^{(T)}(\boldsymbol{z}| \boldsymbol{x})$ ($q^{(T)}( \boldsymbol{\epsilon} | \boldsymbol{z}, \boldsymbol{y})$ samples from HMC), which matches exactly our case.
> > * It is demonstrated in [1] to be superior with respect to other discrepancies in high-dimensional spaces.
> >
> > For further justification, we refer the Reviewer to Table 2 in [2], where authors demonstrated that the usage of this trick leads to higher average test marginal log likelihoods for the MNIST datasets, compared to the case where only the first term of Equation (7) in our paper were considered for the optimization.
> >
> > ### References
> >
> > [1]: W. Gong, Y. Li, and J. M. Hernández-Lobato. Sliced Kernelized Stein Discrepancy. arXiv preprint arXiv:2006.16531, 2020.
> >
> > [2]: A. Campbell, W. Chen, V. Stimper, J. M. Hernandez-Lobato, and Y. Zhang. A gradient Based Strategy for Hamiltonian Monte Carlo Hyperparameter optimization. In International Conference on Machine Learning, pages 1238–1248. PMLR, 2021.

---

> ### Author Response · Authors · 2022-08-09
> **Revised version update: new results demonstrating the significance of our reparameterization method**
>
> We have added incremental improvements after post-rebuttal comments from *Reviewer xZXW* :
>
> * **New experiment on Section A.9 and Figure 10 that demonstrates quantitatively the significance of our novel reparameterization trick as a novel contribution for solving the pathological issues of HMC in hierarchical densities**, in order to address the post-rebuttal concern from *Reviewer xZXW*. This results might also be of interest of *Reviewers jW2a* and *Uewz*.

---

### Meta-Review · Area_Chair_FTyk · 2022-08-27

**Recommendation:** Accept
**Confidence:** Less certain

**Metareview:**

Thanks to the authors for this submission.  The reviewers agreed that this work presented an interesting and novel combination of techniques to achieve good imputation results.  The reviewers also agree that the author-response and subsequent revisions have improved the submission and addressed the vast majority of reviewer concerns — in fact most reviewers increased their scores throughout the process.  I believe this work is technically sound and of interest to the broader community.

**Award:**

No

---

### Decision · Program_Chairs · 2022-09-14

Accept